

# Representing dynamic grass density in the land surface model ORCHIDEE r9010

Siqing Xu[1,2], Sebastiaan Luyssaert[3], Yves Balkanski[1], Philippe Ciais[1,2], Nicolas Viovy[1], Liang Wan[4], Jean Sciare[2]

[1]Laboratoire des Sciences du Climat et de l'Environnement, CEA/CNRS/UVSQ/Université Paris Saclay, Gif-sur-Yvette, France

[2]The Cyprus Institute, Climate and Atmosphere Research Center (CARE-C), Nicosia, Cyprus

[3]Amsterdam Institute for Life and Environment, Department of Ecological Science, Vrije Universiteit Amsterdam, Amsterdam, The Netherlands

[4]Research Institute of Agriculture and Life Sciences, Seoul National University, Seoul, Republic of Korea

*Correspondence to*: Siqing Xu (siqing.xu@lsce.ipsl.fr)





**Abstract.** In semi-arid regions, grasses and shrubs often form spatial heterogeneous patterns interspersed with bare soil, optimizing resource use and productivity. Accurately representing the matrix of vegetation and bare soil in global land surface models is essential for advancing the understanding of the carbon, water, and dust cycles. This study focuses on grasslands

using the land surface model ORCHIDEE (ORganizing Carbon and Hydrology In Dynamic EcosystEms), which originally assumes a globally fixed maximum grass density. This assumption, referred to as the fixed maximum density approach, limits the model's ability to capture grassland responses to environmental changes, resulting in unsustainable productivity and unrealistically frequent mortality events, particularly in resource-limited regions. To address these limitations, we introduced a dynamic density approach that simulates grassland density based on indicators of vegetation growth, such as reserve and

labile carbon content in the grass. The emerging positive correlation between precipitation and simulated grass density supported the validity of the approach. Compared to the fixed maximum density approach, the new approach substantially reduced simulated mortality events, raised the aridity threshold for frequent mortality, and maintained realistic grassland productivity in regions where the presence of grassland is indicated by the observed leaf area index (LAI). This study not only demonstrates that simulating grass density as a function of carbon availability improves ORCHIDEE's capacity to capture

grassland dynamics under environmental variability, but also provides a promising foundation for investigating land–atmosphere feedbacks in (semi-)arid regions.

## 1 Introduction

Grasslands cover up to 40% of the Earth's land surface (Blair et al., 2013) and provide habitats for wildlife and pasture for grazing livestock (Allaby, 2006), through which they contribute to the well-being of more than two billion people (Squires et

al., 2018). The vast distribution of grasslands extends across all ice-free continents, encompassing diverse climatic zones ranging from tropical and temperate to boreal. Major grassland types include prairies, steppes, pampas, velds, and savannas (Allaby, 2006). Among the grassland regions, semi-arid areas are of particular importance due to their role in the global carbon cycle and the turnover of carbon pools (Poulter et al., 2014). In such regions, grasslands are the dominant vegetation, functioning as transitional zones between forests and deserts. They often coexist with shrublands (Smith et al., 2014), deserts

(Cui et al., 2018), or sparsely distributed trees where precipitation is sufficient (Blair et al., 2013; Erdős et al., 2022).

Vegetation structure and density can reveal how plants sustain their communities under extreme environmental conditions. Vegetation patterns in the forms of gaps, labyrinths, strips, and "tiger bush" have been shown to depend on environmental conditions such as humidity, mean temperature, temperature seasonality, and wilting point (Deblauwe et al., 2008). These spatially patterned vegetation structures enhance resource retention and water redistribution, thereby supporting vegetation

survival in dry environments (Galle et al., 1999). Although grasslands are resilient to climate extremes (Erdős et al., 2022; Dodd et al., 2023; Hossain et al., 2023), environmental changes through drought (Ciais et al., 2005), extreme precipitation (Knapp et al., 2008; Craine et al. 2012; Petrie et al., 2018), elevated $CO_2$ (Pan et al., 2022) and heat waves (Karl et al., 1981; Buhrmann et al., 2016; Chang et al., 2020) can affect the growth and survival of grasslands (Toräng et al., 2010; Williams et al., 2007; Prevéy et al., 2015). These impacts are often reflected in grassland density (Ehrlén et al., 2019), particularly under

resource-limited conditions such as water scarcity in (semi-)arid regions (Dyer, 1999). For example, additional winter precipitation promotes grass population growth, while summer drought leads to a decline, highlighting the influence of seasonal precipitation variability on grassland density (Prevéy et al., 2015). Warming was reported to significantly reduce the population of certain plant species in Australian temperate grasslands, indicating a potential warming-driven decline in grassland density (Williams et al., 2007).

In semi-arid regions, when water resources are scarce, grass density declines (Rietkerk et al., 2002), favouring the appearance of bare soil, leading to an enhancement of atmospheric dust emissions (Tegen et al., 2002; Vincenot et al., 2016). Increased dust enhances the summer precipitation due to the absorption of radiation by dust and has been shown to modify the west



African Monsoon pattern (Miller et al., 2014; Balkanski et al., 2021). In contrast, when precipitation is abundant, vegetation grows, bare soil exposure is reduced and soil moisture is increased, which enhances the cohesive forces between soil particles and suppresses dust emissions (Gherboudj et al., 2015). Therefore, the inclusion of dynamic grassland into the land surface models will help to represent these land–atmosphere feedbacks.

Grassland dynamics can be expressed as the variation of "grassland density" or "plant cover". The term "density" in an ecosystem can have multiple definitions, including population density, measured as the number of individuals per area (Zhu et al., 2015), or mass density, expressed as mass per unit area (Rietkerk et al., 2002). In this study, we focus on population density, defined as the number of individuals per unit area, where each individual is assumed to occupy 1 $m^2$ of land. Thus, the unit of grass density in this study is expressed as $m^2$ per $m^2$. For instance, a hectare of grassland with a density of 1 contains 10,000 individuals, covering a total occupied area of 10,000 $m^2$ per hectare (Fig. 1a). A density of 0.25 therefore corresponds to 2,500 individuals covering 2,500 $m^2$ per hectare (Fig. 1b). In this study grass density thus relates to the occupancy of the individuals, and differs from "plant cover" which refers to the projected vegetation coverage in grasslands. Spatially explicit global estimates of grass density remain challenging, due to inconsistent definitions of grass density (Rietkerk et al., 2002) and the inherent difficulties of direct measurement (Vogel et al., 2001; Hamada et al., 2021).

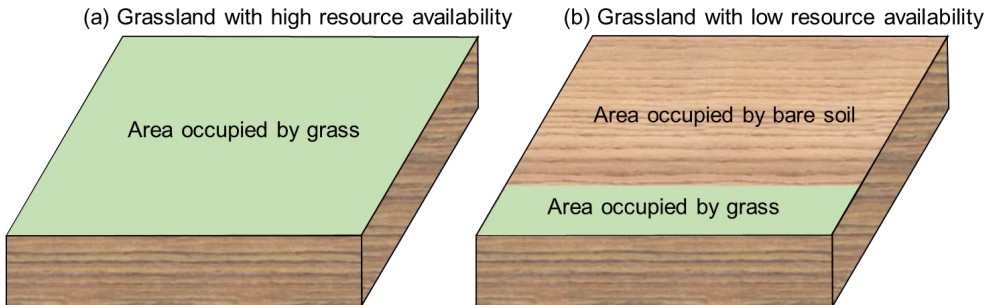

**Figure 1.** Conceptual framework of grassland density under varying resource availability. With high resource availability (**a**), grass density is able to reach the maximum density, while low resource availability (**b**) typically results in lower grass density.

In this study, we use the land surface model ORCHIDEE (trunk version, r9010), which is part of the IPSL-CM Earth System Model (Boucher et al., 2020). Currently, ORCHIDEE prescribes a globally fixed maximum grass density by default, implying that grasslands are stocked with the maximum possible number of individuals. Consequently, the model simulates frequent mortality events driven by this density limit, particularly in (semi-)arid grid cells. Adding to these limitations, a fixed density fails to respond to changes in resource availability, hindering the possibility of studying the response of dust emissions in the presence of grassland when the land surface model is coupled to an atmospheric model (Boucher et al., 2020). Finally, the fixed density renders an evaluation of the model using observed or remotely sensed densities meaningless.

To address the limitations of a fixed grass density in ORCHIDEE, we revised the model to simulate a dynamic grassland density, with the aim of: (a) simulating the response of the bare soil fraction in grasslands to environmental changes, providing a foundation for predicting the long-term spatiotemporal dynamics of dust emissions in future work; (b) enhancing grassland survival, particularly in (semi-)arid regions; and (c) better representing the grassland leaf area index (LAI), which is a key variable in simulating land–atmosphere processes such as photosynthesis, transpiration, albedo and the energy budget. To this aim, we replaced the fixed maximum density approach with a physiology-based approach to simulate grass density in ORCHIDEE, and ran simulations with both approaches to: (1) assess the emergent relationship between grass density and precipitation; (2) compare the frequency of mortality events between the two approaches; (3) quantify the responses of mortality events in grasslands to aridity; and (4) evaluate the simulated leaf area index (LAI) against remote sensing observations.



## 2 Methods

### 2.1 The Land Surface Model ORCHIDEE

ORCHIDEE is a process-based land surface model capable of simulating the carbon, nitrogen, water, and energy cycles, including vegetation dynamics, biogeochemical fluxes, and plant competition (Krinner et al., 2005; Naudts et al., 2015; Vuichard et al., 2019). In ORCHIDEE trunk version r9010, each grid cell may contain up to 15 plant functional types (PFTs), representing eight different types of forests, four types of grasslands, two types of croplands, and bare soil (defined as a separate PFT). Each PFT has a value for its fraction ($V_{fra}$), and the sum of $V_{fra}$ from all 15 PFTs is equal to 1 within each grid cell. The value of $V_{fra}$ is derived from a land cover map that is in turn obtained from post-processing of remote sensing observations (Poulter et al., 2015; ESA, 2017).

ORCHIDEE distinguishes four grassland types: temperate $C_3$ grassland, tropical $C_3$ grassland, boreal $C_3$ grassland, and $C_4$ grassland. $C_3$ and $C_4$ plants use the $C_3$ or $C_4$ photosynthesis pathways, respectively (Taylor et al., 2010). In the light of this study's emphasis on (semi-)arid grasslands, boreal $C_3$ grassland was excluded from the analysis. For each PFT, ORCHIDEE simulates processes such as photosynthesis, phenology, carbon and nitrogen allocation, senescence, turnover, and mortality, based on PFT-specific parameter sets.

### 2.2 Grass density

#### 2.2.1 Fixed maximum density approach

The grassland density in ORCHIDEE is calculated as:

$$D = N_{max} \tag{1}$$

where $D$ refers to grass density, defined as the fraction of area occupied by individuals within a unit of grassland area ($m^2$ per $m^2$). By default, the maximum number of individuals ($N_{max}$) in grassland is set to 10,000 per hectare. In the model, each individual is assumed to occupy 1 $m^2$. As a result, this leads to a fixed grass density of 1 $m^2$ per $m^2$.

#### 2.2.2 Dynamic density approach

A grassland with the abundant resources (i.e., water, light, nitrogen, temperature, and $CO_2$) is able to support high grass density (Fig. 1a) with high biomasses per individual. In contrast, for grasslands with limited resources, both the density and the individual biomass might be low (Fig. 1b). In the fixed maximum density approach, for grasslands with limited resources, the available resources per individual might be too limited to sustain sufficient reserve and labile carbon, thereby limiting vegetation growth. In an individual plant, carbon biomass is distributed among different pools, including leaf, aboveground stem, root, fruit, reserve and labile. Carbon in the leaf, aboveground stem, root, and fruit pools is allocated to specific structural components, whereas carbon in the reserve and labile pools is non-structural and can be found throughout the plant. Reserve carbon refers to a more stable pool, whereas labile carbon represents a rapidly mobilized pool (Gupta and Kaur, 2000). Reserve and labile carbon are considered primary sources for immediate regrowth following defoliation or the alleviation of stress (Volaire, 1995). In this study, we considered the sum of reserve and labile carbon for subsequent calculations.

The dynamic density approach simulates grassland density by adjusting the population to the maximum number of individuals that can be sustained by the available resources. This adjustment is performed on a daily basis, primarily in response to the total reserve and labile carbon levels.

When the reserve and labile carbon in the plant drop below their respective target values and this condition persists over a longer time period, a mortality risk ensues (Volaire, 1995). The dynamic density approach in ORCHIDEE accounts for this case by decreasing the number of individuals when the total reserve and labile carbon fall below a simulated target value. Therefore, the labile and reserve carbon are redistributed among the fewer plants, whose chances for future survival have



increased. Note that the carbon of other compartments (including leaf, aboveground stem, root and fruit) in each individual
remains constant when the number of individuals is decreased. The nitrogen content of each compartment in an individual is
updated by multiplying the previous nitrogen content by the ratio of the previous density to the current density. A minimum
density of 0.05 is set to avoid numerical errors in the model.

The ORCHIDEE model distinguishes eight phenological stages ($P_S$): (1) Planting: emergence of new plant, (2) Buds:
appearance of buds, (3) Leaf: onset of leaf, (4) Growth: presence of canopy, (5) Pre-senescence: cessation of vegetation growth,
(6) Senescence: senescence of plant, (7) Dormancy: dormancy of plant and (8) Death: death of plant. It has been reported
previously (e.g. Volaire, 1995; Sarath et al., 2014; Keep et al., 2021) that during vegetation senescence, the reserve or labile
carbon experiences a peak influx of carbon reallocated from leaves, and during dormancy, reserves are conserved for the
upcoming growing season. Therefore, grassland density is only decreased during the following phenological stages: pre-
senescence, senescence, and dormancy, when the reserve and labile carbon should attain their annual peak. A decrease in
grassland density in one timestep of the model is calculated as follows:

$$
\begin{cases}
C_{\text{all}} \times D_1 = \left(C_{\text{all}} - C_{\text{res},1} - C_{\text{lab},1}\right) \times D_2 + T_{\text{res}} + T_{\text{lab}} \\
D_2 = \frac{\left(C_{\text{all}} \times D_1 - T_{\text{res}} - T_{\text{lab}}\right)}{\left(C_{\text{all}} - C_{\text{res},1} - C_{\text{lab},1}\right)}
\end{cases}
\tag{2}
$$

where $C_{\text{all}}$, $C_{\text{res}}$, and $C_{\text{lab}}$ are the carbon biomasses of all compartments, the reserve, and the labile pool of an individual,
respectively (gC per individual); the indices 1 and 2 refer to the value before and after the density adjustment; $T_{\text{res}}$ and $T_{\text{lab}}$ are
the targets for reserve carbon and labile carbon in the PFT (gC m$^{-2}$), respectively, which are calculated in ORCHIDEE as:

$$
T_{\text{res}} = \min\left[\beta \times \left(B_{\text{root}} + B_{\text{stem,above}} + B_{\text{stem,below}}\right), \frac{\exp(\text{LAI} \times k) - 1}{k \times SLA_{init}} \times \left(1 + \frac{\delta}{\lambda}\right)\right]
\tag{3}
$$

$$
T_{\text{lab}} = \max\left[t \times \gamma \times \left(N_{\text{leaf}} + N_{\text{root}} + N_{\text{fruit}} + N_{\text{stem,above}} + N_{\text{stem,below}}\right), 10 \times GPP_{week}\right]
\tag{4}
$$

where $\beta$ is the fraction of stem mass stored in the reserve pool during the growing season (unitless); $B_{\text{root}}$, $B_{\text{stem,above}}$, and
$B_{\text{stem,below}}$ are the carbon biomasses of the root, aboveground stem, and belowground stem, respectively, for a given PFT (gC
m$^{-2}$); $k$ is the extinction coefficient of the leaf nitrogen content profile within the canopy (unitless); $SLA_{init}$ is the initial specific
leaf area at the bottom of the canopy (m$^2$ gC$^{-1}$); $\delta$ is the fraction of maximum root biomass covered by reserve biomass (unitless);
$\lambda$ is a scaling factor converting stem mass into root mass (unitless); $t$ is the turnover coefficient of the labile carbon pool
(unitless); $\gamma$ is the parameter used to calculate the labile pool (unitless); $N_{\text{leaf}}$, $N_{\text{root}}$, $N_{\text{fruit}}$, $N_{\text{stem,above}}$ and $N_{\text{stem,below}}$ are the nitrogen
biomasses of the leaf, root, fruit, aboveground stem, and belowground stem in the PFT, respectively (gN m$^{-2}$); and $GPP_{week}$ is
the weekly gross primary productivity (gC m$^{-2}$ per day).

When the carbon content in both reserve and labile pools exceeds their respective targets, and carbon is present in the fruit
pool, grass density will be increased. The excessive carbon from the fruit, reserve, and labile pools will be used to create new
individuals. After updating the number of individuals, the carbon in labile and reserve pools is reset to their target values, and
the carbon in fruit pool becomes zero. The density is only increased during the phenological stage labelled as "Growth". This
approach for increasing grassland density reflects grass recruitment through asexual means, which is a suitable method for
representing perennial plants (Blair et al., 2013). Note that the carbon of other compartments (including leaf, aboveground
stem and root) in each individual remains constant. Nitrogen is treated using the same method as that applied to decreasing
density. An increase in grassland density is calculated as follows:

$$
\begin{cases}
C_{\text{all}} \times D_1 = \left(C_{\text{all}} - C_{\text{res},1} - C_{\text{lab},1} - C_{\text{fruit},1}\right) \times D_2 + T_{\text{res}} + T_{\text{lab}} \\
D_2 = \frac{\left(C_{\text{all}} \times D_1 - T_{\text{res}} - T_{\text{lab}}\right)}{\left(C_{\text{all}} - C_{\text{res},1} - C_{\text{lab},1} - C_{\text{fruit},1}\right)}
\end{cases}
\tag{5}
$$



where $C_{fruit,1}$ refers to the carbon biomass from the fruit pool at the individual level before density adjustment (gC per individual).

### 2.3 Determination of PFT Dominance and Co-existence

We used the fraction of vegetation type ($F_{ra}$), defined as the ratio of the area covered by a given PFT to the total area of the grid cell, as the basis for classification. We defined a grassland PFT as "dominant" in a given grid cell if two conditions were simultaneously met: (1) its $f_{ar}$ was 0.5 or higher, and (2) the $V_{fra}$ of other grassland PFTs was lower than 0.1. When two or more grassland PFTs had $V_{fra}$ values above 0.1, they were considered to co-exist.

### 2.4 Mortality events in grasslands

In reality, under very arid conditions, plants may be unable to grow during unfavourable years (Blair et al., 2013; Bodner et al., 2017). However, seed banks or rhizomes in the soil can enable regrowth when environmental conditions become favourable again (Blair et al., 2013). In ORCHIDEE, transferring carbon from reserve to leaves initiates the growing season for deciduous vegetation, but it does not account for a long-term reserve pool that can retain reserves over several years. Under unfavourable conditions, the reserves are rapidly depleted, preventing vegetation from re-growing in the following years. In ORCHIDEE, this can be monitored through phenological stages ($P_S$), which is set to one of the eight stages of the plant (see above).

When, at the end of the growing season, the reserve and labile carbon pools are empty, grassland dies in the ORCHIDEE model and $P_S$ is set to "Death". At the start of the next growing season, a new grassland is artificially replanted by the model, and its $P_S$ is updated to "Planting". The initial biomass from the artificially replanted grassland is taken directly from the atmosphere with a fixed value, thus bypassing germination and the early stages of plant development. The transition of the phenological stages from "Death" to "Planting" is recorded and counted as a single mortality event. If, in the meantime, the environmental conditions become favourable again, the grassland will continue to grow. If not, it will die again. Therefore, the vegetation in each grid cell might experience repeated mortality events over decades, as it is replanted following each event (Fig. S1).

As long as these mortality events remain infrequent, they have little impact on the simulated fluxes because the cumulated photosynthesis by far exceeds the carbon used to initialize the grassland biomass. If, however, the mortality and replanting cycles become frequent, the repetitive addition of the initial carbon may become substantial compared to carbon assimilation through photosynthesis. For example, in tropical $C_3$ grassland at grid cell (59° E, 39° N) using the fixed maximum density approach, frequent mortality events resulted in unrealistic productivity patterns, characterized by abrupt post-mortality spikes resulting from the addition of the initial biomass when replanting (Fig. S1).

Grasslands are modelled as perennial vegetation in ORCHIDEE; hence, a skilful model is expected to simulate infrequent mortality events in grasslands, reflecting the inherent resilience and stability of grassland ecosystems. We defined a threshold for such "infrequent mortality" as fewer than five times over a 51-year simulation period. This benchmark was supported by ecological records of prolonged droughts, such as the 1930s Dust Bowl in the Great Plains (Blair et al., 2013) and a decade-long drought in southern Arizona (Bodner et al., 2017). Both of these events caused widespread mortality in perennial grasslands, implying that mortality on a decadal scale (i.e., approximately once per decade) is realistic under extreme conditions. Consequently, exceeding such a frequency might indicate unrealistic model behaviour. Therefore, reducing the number of mortality events is considered an indication of model improvement for the representation of grasslands.

### 2.5 Leaf area index (LAI) for grasslands

Remote sensing-based LAI observations (Myneni et al., 2021; Wan et al., 2024) were used to evaluate model output of LAI for grasslands. We utilized two observational (satellite-based) datasets: MODIS LAI (2004–2020), originally at a resolution



of 1 km with a 4-day temporal frequency (Myneni et al., 2021), and Sentinel-2 data for 2019 at a spatial resolution of 10 m (Wan et al., 2024). In order to compare them with the 2° × 2° global simulations in ORCHIDEE, both datasets were aggregated
to the same spatial resolution. In both cases, LAI values were filtered based on land cover classification, to retain only grassland pixels. For MODIS, the grassland classification was based on MCD12Q1 land cover product (Myneni et al., 2021). The resulting grassland grid cells were then spatially regridded using a first-order conservative remapping for grid cells classified as grassland, which preserves the total integrated quantity during spatial interpolation, implemented via RemapCon (Jones, 1998; Goudiaby et al., 2024) in the Climate Data Operators (CDO) for Linux. For Sentinel-2, grassland pixels were identified
using the GLC_FCS30D land cover dataset (Zhang et al., 2024). The LAI values were aggregated by computing the arithmetic mean of all high-resolution pixels classified as grassland (code 130) within each 2° × 2° grid cell, using Python scripts.

Leaf area index is a central variable in climate models as it is one of the functional links between the land surface and the atmosphere. Thus, a skilful land surface model is expected to simulate realistic spatial and temporal patterns in LAI. In principle, LAI is the product of leaf carbon mass and specific leaf area (SLA). In ORCHIDEE, SLA is not assumed to be
constant but varies vertically through the canopy. To account for this vertical variation, LAI is calculated using a dynamic SLA formulation, as expressed in Eq. (6), where the extinction coefficient $k$ represents the vertical profile of leaf nitrogen content within the canopy. This calculation is performed at the PFT level as follows:

$$\text{LAI} = \frac{\log(1+ k \times B_{\text{leaf}} \times SLA_{\text{init}})}{k} \tag{6}$$

where $B_{\text{leaf}}$ refers to the leaf biomass (gC m$^{-2}$) calculated as the product of individual leaf biomass (gC per individual) and
grass density.

In ORCHIDEE, land cover map was derived from the ESA CCI Land Cover dataset (ESA, 2017) and converted into PFT fractional maps using a cross-walking table (Poulter et al., 2015). However, neither observational dataset distinguishes specific types of grasslands. To enable comparison, all grassland PFTs in ORCHIDEE were merged into a single grassland category by computing the weighted average LAI across PFTs, using the fraction of vegetation type ($V_{\text{fra}}$) for each grassland PFT as the
weighting factor. Only grid cells where the total $V_{\text{fra}}$ of grassland PFTs exceeded 0.1 in ORCHIDEE were included for the LAI analysis.

## 2.6 Aridity

Aridity ($A$) serves as an important indicator of water scarcity (Berdugo et al., 2020), and is calculated as:

$$A= 1 - \frac{P}{E_{\text{PET}}} \tag{7}$$

where $P$ is precipitation, taken from input forcing data, and $E_{\text{PET}}$ refers to potential evapotranspiration simulated in ORCHIDEE. A high aridity value, e.g., 0.5 or more, indicates that the region is limited in water resources.

## 2.7 Tuning of C$_4$ grassland parameters

Field observations indicate that C$_4$ grasslands are generally more drought-tolerant than C$_3$ grasslands (Taylor et al., 2010). However, in ORCHIDEE, the simulated density of C$_4$ grasslands was too high under strong water limitations (Fig. S2; Fig.
S3a; Fig. S4b). This suggests that the model underestimates the sensitivity of C$_4$ grasslands to water stress, leading to an overestimation of grass density in these areas. This issue was addressed by recalibrating PFT-specific parameters for C$_4$ grasslands, related to the targets for reserve and labile carbon (see below) and the water stress function (see below). The recalibrated parameters were retained as they improved the correlation between precipitation and grass density (Fig. S3).

The first parameter that was recalibrated controls the target level for reserve and labile carbon, which is a critical driver of
grass density in the dynamic density approach. This adjustment was tested in Southern Africa for C$_4$ grasslands, by applying



a scaling factor to reserve and labile carbon target, where values of 0.5 and 1.5 (Fig. S4c and d) represent lower and higher carbon targets, corresponding to less and more stringent growth requirements, respectively. In this study, a value of 1.5 was applied to increase the target level for reserve and labile carbon in $C_4$ grasslands.

The second parameter controls the water stress for transpiration, which ranges from 1 (no stress) to 0 (severe stress). By default, the function of water stress ($W_{stress}$) for transpiration is formulated in the linear form as follows:

$$W_{stress} = \min\left(1, \max\left(0, \frac{\theta - \theta_{wilt}}{\theta_{nostress} - \theta_{wilt}}\right)\right) \times R_{profile} \tag{8}$$

where $\theta$ refers to soil moisture (kg m$^{-2}$), $\theta_{wilt}$ refers to soil moisture at the wilting point (kg m$^{-2}$), $\theta_{nostress}$ is the soil moisture when there is no water stress (kg m$^{-2}$), and $R_{profile}$ is the normalized root mass or length fraction (unitless.)

This water stress function can be switched to an exponential formulation with a parameter $\alpha$., written as below:

$$W_{stress} = \min\left(1, \exp\left[-\alpha \times \left(\frac{\theta_{fc} - \theta_{wilt}}{\theta_{nostress} - \theta_{wilt}}\right) \times \left(\frac{\theta_{nostress} - \theta}{\theta - \theta_{wilt}}\right)\right]\right) \times R_{profile} \tag{9}$$

where $\theta_{wilt}$ refers to soil moisture at field capacity (kg m$^{-2}$).

This $\alpha$ value can range from 0 to 10, with higher values indicating the plant is more limited by water stress (Raoult et al., 2021). In this study, values of $\alpha$ = 1, 2, 4, and 8 were tested for Southern Africa (Fig. S5), and $\alpha$ = 8 was selected for the global simulations to enhance water stress sensitivity in $C_4$ grasslands.

## 2.8 Simulations

All simulation experiments started with a 200-year spin-up at a spatial resolution of 2° × 2° at a global scale while cycling over the CRU-JRA climate forcing from 2004 to 2020 (Harris et al., 2020), which aligns with the period of MODIS LAI observations (Fig. 2). In these simulations, the $CO_2$ concentration was set to 350 ppm globally, and the land cover map was for the year 2004, derived from ESA CCI Land Cover dataset (Poulter et al., 2015; ESA, 2017). The year 2004 was chosen as it corresponds to the starting year of the MODIS data for LAI used in this study. Hence, land cover change is not accounted for in this experiment. During the spin-up, the soils are brought to equilibrium every 15 years with the 15-year average litter inputs. To account for the nitrogen cycle in the ORCHIDEE model, the semi-analytical spin-up process has to be repeated 10 times before an equilibrium is reached (Vuichard et al., 2019). After 150 years of simulation, we checked that the soil passive carbon pools had reached an equilibrium, with the values fluctuating within a 20% range over the 51 simulation years (Lardy et al., 2011). Once the soil carbon and nitrogen pools were in equilibrium, the simulation experiment branched off with two configurations using the fixed maximum density approach (Sect. 2.2.1), and the dynamic density approach (Sect. 2.2.2).

The simulation results from the two configurations were analysed by comparing the grass density, mortality, the response of mortality in grassland to aridity, and the mean annual grassland LAI. To align with the 17-year period (2004–2020) of the CRU-JRA forcing data, we present below the averaged values for grassland density and mean annual LAI over the same period. The mortality events were accumulated and analysed over 51 simulation years taken after the model had reached equilibrium following 150 years of simulation.



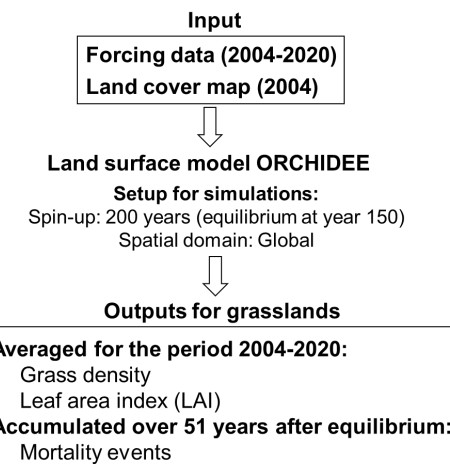

**Figure 2.** Overview of ORCHIDEE model inputs, setup, and outputs.

## 3 Results

### 3.1 Spatial distribution of simulated grass density

With the dynamic density approach, the globally simulated grass density varied between the minimum and maximum threshold values of 0.05 and 1, respectively (Fig. 3). Throughout the 17-year period (2004–2020), 56% of the grid cells in temperate $C_3$ grasslands, 66% in the $C_4$ grasslands, and 33% in the tropical $C_3$ grasslands maintained the maximum density (Fig. 3a–c). In contrast, grassland density reached the minimum value in fewer than 1% of the grid cells across all three grassland types. The majority of grassland grid cells had a density ranging between 0.9 and 1, i.e., 75% of temperate $C_3$ grasslands, 79% of $C_4$

grasslands, and 59% of tropical $C_3$ grasslands (Fig. 3d–f).

Temperate $C_3$ grasslands in eastern USA, Europe, eastern China, and New Zealand were simulated at the maximum density, in contrast to areas such as western USA, Middle East and northern China where ORCHIDEE simulated lower densities. The $C_4$ grasslands were simulated at a lower than maximum density in India, Sahel, southern Africa and middle Australia. By contrast, in tropical $C_3$ grasslands, densities between the minimum and the maximum were simulated in regions such as the

Middle East, northern Africa, southern Australia, and southern Africa, whereas the maximum density was simulated in South America and southern east Asia.



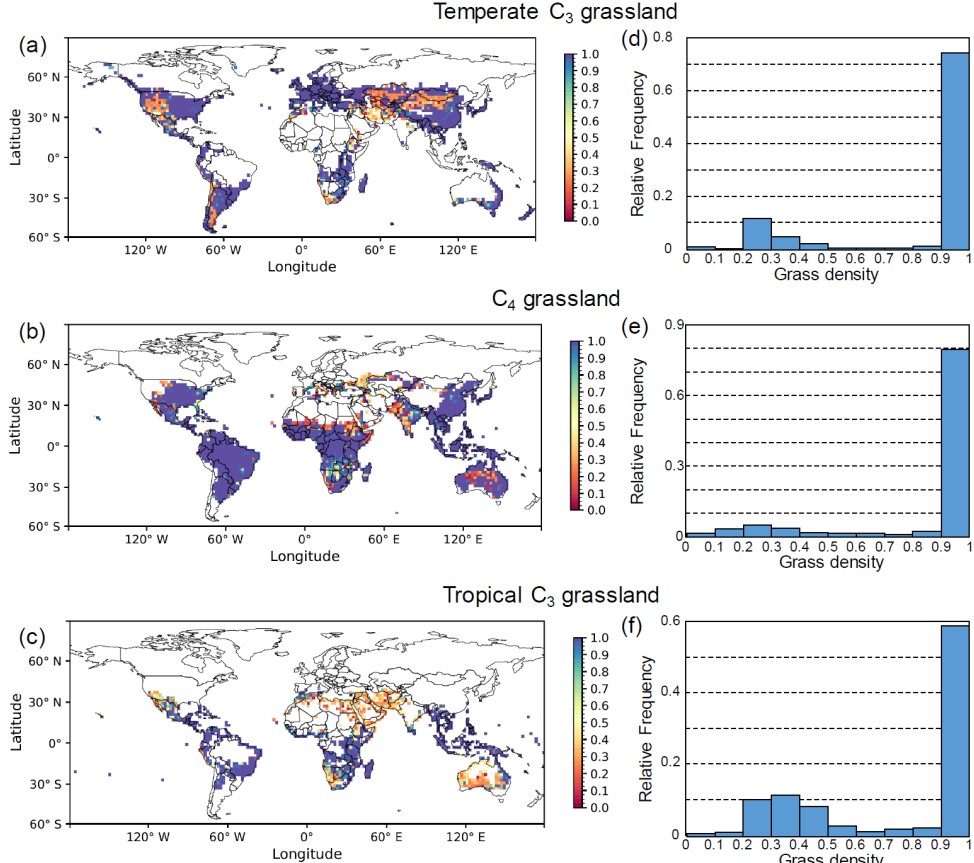

**Figure 3.** Global distribution (**a–c**) and frequency histograms (**d–f**) of simulated grass density for three grassland types. Global maps show the 17-year average (2004–2020) grass density simulated with the dynamic density approach for (**a**) temperate $C_3$, (**b**) $C_4$, and (**c**) tropical $C_3$ grasslands. The corresponding histograms (**d–f**) show the relative frequency of simulated density values for each grassland type.

### 3.2 Plausibility check of simulated grassland density based on vegetation composition and precipitation

As a reminder for the following results, the land cover map represents the fraction of vegetation type ($V_{fra}$) for each PFT within

one grid cell, whereas grassland density reflects grass and bare soil fractions within the grassland PFT. A high $V_{fra}$ for a given PFT indicates that the PFT is the dominant vegetation type in that area, but it does not necessarily mean that the PFT has a high vegetation density. However, under natural and undisturbed conditions, our expectation is that when a PFT is dominant (explained in Sect. 2.3), its grass density should be higher than that of other PFTs with lower $V_{fra}$. Conversely, when multiple PFTs coexist (Sect. 2.3), their densities should be similar. Therefore, we calculated the relative contribution of each of the

three grassland types to the total grass density (Fig. 4b), allowing us to focus on their proportional importance rather than their absolute values.

The land cover map for each grassland type (Fig. S6) provides information on $V_{fra}$. Based on this, grassland categories (Fig. 4a) were classified using the $V_{fra}$ thresholds introduced in Sect. 2.3. Grid cells with a total $V_{fra} \geq 0.1$ for all grassland types are marked in light grey, representing regions of significant grassland presence. Within these areas, seven distinct grassland

categories were identified. Specifically, among the identified cells, 3% of grid cells were dominated by temperate $C_3$ grasses (red), 5% by $C_4$ grasses (green), and 0.4% by tropical $C_3$ grasses (blue). Co-existence patterns included: temperate $C_3$ and $C_4$ grasses (12%, yellow), temperate and tropical $C_3$ grasses (1%, purple), tropical $C_3$ and $C_4$ grasses (6%, cyan), and all three grassland types co-existing (0.7%, dark grey).





Figure 4b shows ternary plots illustrating the grass density composition among the three grassland types. In these plots, the
position of each point indicates the relative proportion of grass density from the three grassland PFTs within that grid cell:
temperate C$_3$ (top apex), C$_4$ (bottom left), and tropical C$_3$ (bottom right). To achieve this, the densities of the three PFTs in
each grid cell were normalized such that their sum equals 1, thereby highlighting their relative contributions. Point size
corresponds to the frequency of similar compositions across grid cells, with larger points indicating higher occurrence. As
expected, points tended to cluster near the apex corresponding to the dominant PFT in grid cells where a single grassland type
prevailed, reflecting a higher proportional contribution of that PFT to overall grass density. Deviations occurred where non-
dominant PFTs contributed to grass density but appeared with lower frequency. In cases of two-PFT coexistence, points were
predominantly distributed along the edges between the respective apices, while in three-PFT coexistence scenarios, points
shifted toward the centre of the triangle, indicating a more balanced contribution of grass density among the three grassland
types.

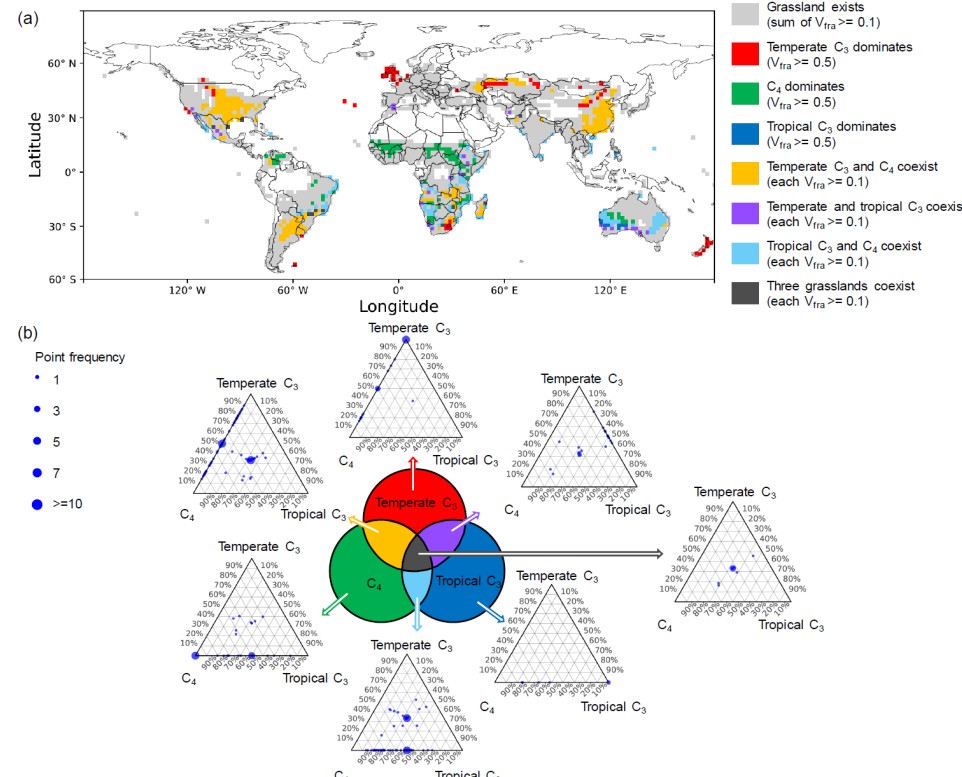


**Figure 4.** Global distribution of grassland categories and their relative grass density composition. (**a**) Grassland classification based on
fraction of vegetation type ($V_{fra}$). Grey areas indicate where the total grassland $V_{fra} \geq 0.1$. These areas are further classified into seven
categories: (1) Temperate C$_3$-dominated ($V_{fra}$ of temperate C$_3$ grassland $\geq 0.5$ while $V_{fra}$ in other grassland PFTs <0.1, red); (2) C$_4$-dominated
($V_{fra}$ of C$_4$ grassland $\geq 0.5$ while $V_{fra}$ in other grassland PFTs <0.1, green); (3) Tropical C$_3$-dominated ($V_{fra}$ of tropical C$_3$ grassland $\geq 0.5$
while $V_{fra}$ in other grassland PFTs <0.1, blue); (4) Temperate C$_3$ and C$_4$ co-existence ($V_{fra} \geq 0.1$ for both types while $V_{fra}$ in tropical C$_3$
grassland < 0.1, yellow); (5) Temperate and tropical C$_3$ co-existence ($V_{fra} \geq 0.1$ for both types while $V_{fra}$ in C$_4$ grassland < 0.1, purple); (6)
Tropical C$_3$ and C$_4$ co-existence ($V_{fra} \geq 0.1$ for both types while $V_{fra}$ in temperate C$_3$ grassland < 0.1, cyan); (7) Three grassland types co-
existence ($V_{fra} \geq 0.1$ for all grassland types, dark grey). (**b**) Ternary plots showing the relative proportions of normalized grass density for
the three grassland types (temperate C$_3$, C$_4$ and tropical C$_3$) within each grid cell. Each subplot corresponds to one of the seven grassland
categories in (**a**). Point size represents the frequency of occurrence at a given composition.

Direct comparison with globally observed grassland density is limited due to measurement challenges (Vogel et al., 2001;
Hamada et al., 2021) and inconsistent definitions of "grassland density". As an alternative to direct comparison, the plausibility
of the simulated grassland densities was assessed by analysing the relationship between precipitation and grassland density.




Two-dimensional kernel density estimation (2D KDE) plots (Fig. 5) illustrated the distribution of grass density along the
precipitation gradient for three grassland types. Grass density exhibited a positive correlation with precipitation for all three
grassland types, forming two distinct clusters separated by a threshold of approximately 0.7. Above this value, grass density
increased with precipitation. The peak probability density (indicated in yellow) for all grassland types occurred at grass density
values between 0.9 and 1, and at precipitation rates ranging from 0.5 to 4 mm per day. When grass density was below 0.7, a
positive trend with precipitation remained evident, particularly for tropical $C_3$ grasslands.

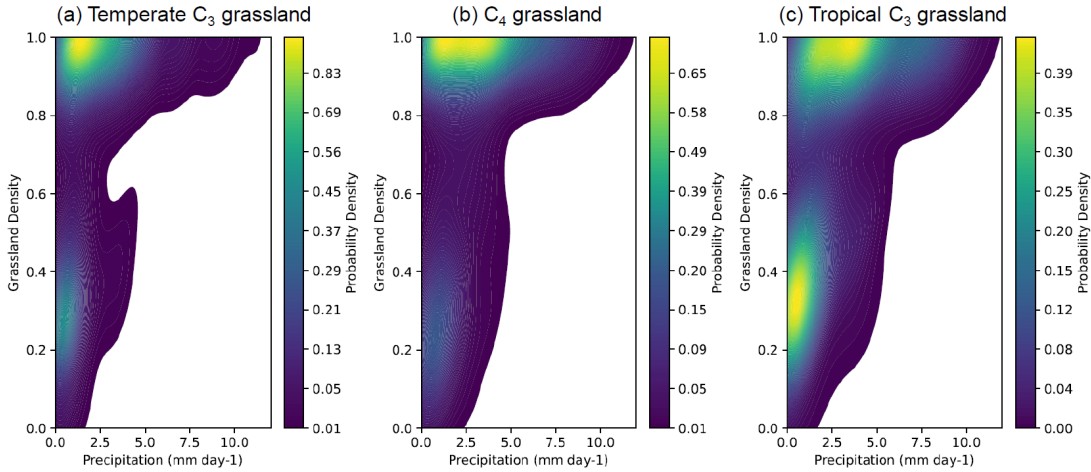

**Figure 5.** Relationship between precipitation and grass density. The two-dimensional kernel density estimation (2D KDE) plots illustrate
the correlation between precipitation and grass density for temperate $C_3$ (**a**), $C_4$ (**b**), and tropical $C_3$ (**c**) grasslands. Lighter colours indicate
higher probability densities, while darker colours represent lower probability densities. Grass density and precipitation values were averaged
over the period 2004–2020.

**3.3 Mortality events simulated in grasslands**

The fixed maximum density approach, used in this study as the reference, prescribes the grass density at unity, implying
grasslands are at their maximal density regardless of whether the environmental conditions are favourable. The grassland
survival in the model depends on productivity per individual, which is represented as the gross primary productivity (GPP) per
individual minus artificially injected carbon from replanting events. When this value drops below an arbitrary threshold of $10^{-4}$ gC per individual, ORCHIDEE considers it insufficient for grassland survival and kills the vegetation. Over the 17-year
period (2004–2020), in the fixed maximum density approach, 40%, 32%, and 58% of grid cells in temperate $C_3$, $C_4$, and tropical
$C_3$ grasslands, respectively, fell below this critical productivity threshold albeit with different frequencies (Fig. S7a–c). By
implementing the dynamic density approach, these proportions decreased to 24%, 26%, and 34%, respectively (Fig. S7d–f).

Over 51 simulation years, the mortality of temperate $C_3$ grasslands with the dynamic density approach was reduced in 98%
(Fig. 6d) of the grid cells when compared to the fixed maximum density approach (Fig. 6a). Similarly, $C_4$ grasslands (Fig. 6e)
and tropical $C_3$ grasslands (Fig. 6f) showed a reduction in 97% and 99% of the grid cells, respectively. For the dynamic density
approach, 51% of the grid cells in temperate $C_3$ grasslands experienced zero mortality events over the 51-year simulation
period, compared to 55% for $C_4$ grasslands and 50% for tropical $C_3$ grasslands. The dynamic density approach helped address
the high frequency of mortality events simulated with the fixed maximum density approach (Fig. 6a–c), with paired t-tests
showing a significant reduction in the mortality for temperate $C_3$ grasslands (n = 1526 grid cells, $p < 0.001$), $C_4$ grasslands (n
= 1776 grid cells, $p < 0.001$), and tropical $C_3$ grasslands (n=1001 grid cells, $p < 0.001$).



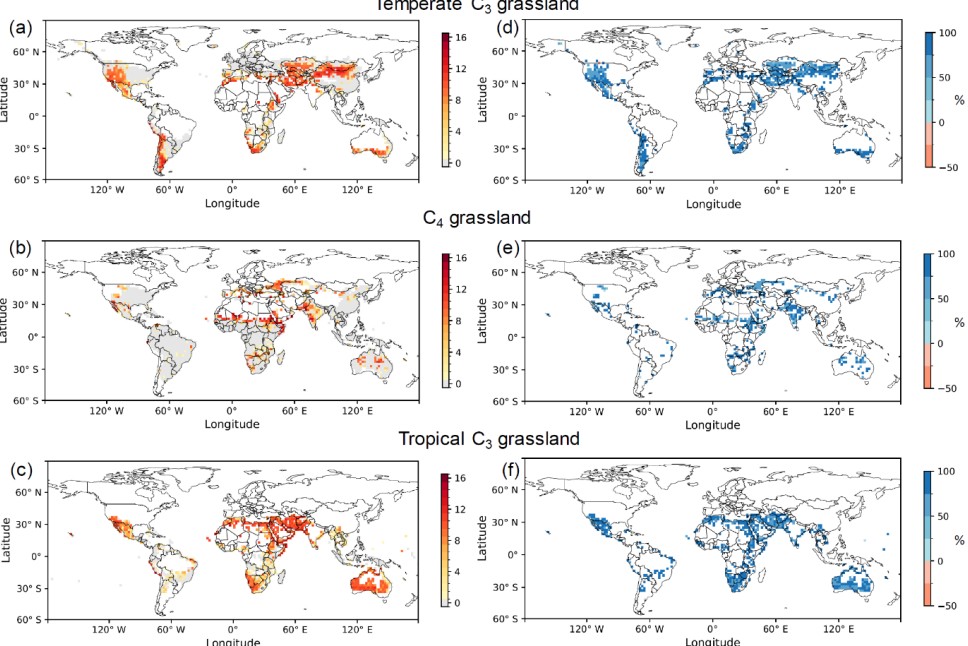

**Figure 6.** Counts of mortality events using the fixed maximum density approach (**a–c**) and the relative mortality reduction using the dynamic density approach (**d–f**) for temperate $C_3$ grasslands (**d**), $C_4$ grasslands (**e**), and tropical $C_3$ grasslands (**f**). The number of mortality events was accumulated over a 51-year test simulation, driven by climate forcing data from 2004 to 2020 and conducted after a 150-year spin-up period that allowed the model to reach equilibrium. The relative reduction was calculated by subtracting the number of mortality events in the dynamic density approach from those in the fixed maximum density approach, and then dividing the result by the number of events in the fixed approach. Positive values (in blue) signify that the dynamic density approach reduced mortality, while negative values (in red) signify an increase.

### 3.4 Relationship between aridity and mortality events on grasslands

We examined the correlation between aridity and mortality events for both density approaches over a 51-year simulation driven by climate forcing data from the 2004–2020 period. In the fixed maximum density approach, in which grass density does not respond to resource availability, mortality events occurred within an average aridity range of 0.3 to 1 (Fig. 7a–c). This range was especially wide for temperate and tropical $C_3$ grasslands, whereas for $C_4$ grasslands, the aridity range was restricted to values between 0.6 and 1 (Fig. 7a–c). For the dynamic density approach (Fig. 7d–f), mortality occurred at a higher aridity

compared to the fixed maximum density approach (Fig. 7a–c). This was illustrated by temperate $C_3$ grasslands, for which frequent mortality events (assumed to occur 5 times or more over 51 years, Sect. 2.4) occurred at an aridity of at least 0.3 in the fixed maximum density approach (Fig. 7a), and at an aridity greater than 0.9 in the dynamic density approach (Fig. 7d). These thresholds were slightly higher for $C_4$ grasslands at 0.7 (Fig. 7b) and 0.8 (Fig. 7e), while for tropical $C_3$ grasslands, the thresholds were 0.4 (Fig. 7c) and 0.9 (Fig. 7f) for the fixed maximum density and dynamic density approaches, respectively.

The dynamic approach effectively suppressed mortality events, particularly under highly arid conditions. For instance, when aridity exceeds 0.9, fewer than seven mortality events were simulated in the dynamic density approach (Fig. 7d–f). To quantify the overall impact, we aggregated mortality events for 51 simulation years across all grassland types and grid cells. When aridity was lower than 0.9, the total number of mortality events decreased by a factor of 7, from 1,642 (fixed maximum density approach) to 228 (dynamic density approach). This reduction reached a factor of 5 under higher aridity (aridity ≥ 0.9), with

the event count decreasing from 10,533 to 2,247.



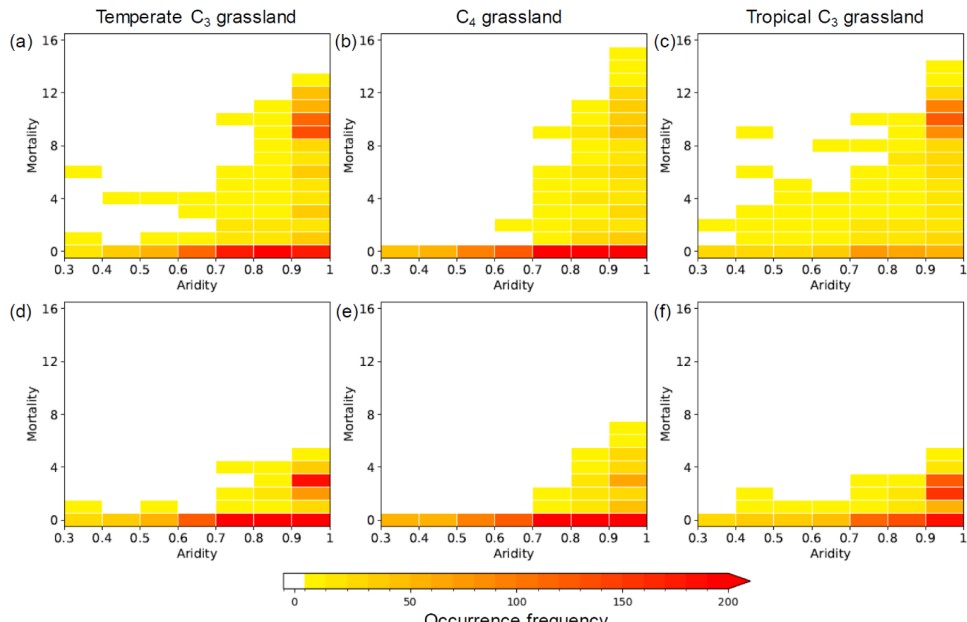

**Figure 7.** Relationship between aridity and mortality events over three types of grassland. Panels (**a–c**) show the relationship using the fixed maximum density approach, while panels (**d–f**) show it using the dynamic density approach. The grassland types are temperate C3 (**a, d**), C4 (**b, e**), and tropical C3 (**c, f**). The mortality events were accumulated over 51 simulation years, and the aridity was calculated for the same period.

### 3.5 Comparison between simulated and observed LAI in grasslands

In this section, we assessed the simulated grassland LAI not merely as a value for direct comparison with observations, but more importantly, as a crucial indicator of improved ecosystem function under the dynamic density approach—specifically, its ability to sustain productivity and reduce mortality.

MODIS remote sensing observations were used to determine the presence of grasslands. Grasslands were considered present in pixels where the MODIS vegetation mask indicated that grassland should be present and where the observed LAI was greater than 0.1 (Tian et al., 2004; Hajj et al., 2014; Lawal et al., 2022). For 60% of the grid cells identified by MODIS as grasslands, ORCHIDEE simulated grassland with a LAI > 0.1 in both approaches. Among these grid cells, 64% showed no mortality events with the fixed maximum density approach. This proportion increased to 86% with the dynamic density approach. Furthermore, among the grid cells that exhibited mortality with the fixed maximum density approach, 97% showed a reduction in mortality in the dynamic density approach.

In both approaches, simulated mean annual LAIs were generally lower than observed LAIs in grasslands (Fig. 8a). This underestimation was especially pronounced in South Asia, South America, and Central Africa, with differences ranging from –0.5 to –3 (Fig. 8b, c). Conversely, simulated LAIs were higher than observations (differences of +0.5 to +3) mainly in southwest China and Mongolia. In Australia and Southern Africa, the differences were less pronounced. Furthermore, grassland LAI from Sentinel-2 data (2019) was consistent with the LAI simulated by the dynamic density approach in Southern Africa and Australia (Wan et al., 2024; Fig. S8). Compared to the fixed maximum density approach, the dynamic density approach resulted in a higher mean annual LAI for 26% of the grid cells and a lower LAI for the remaining 74% (Fig. 8d).





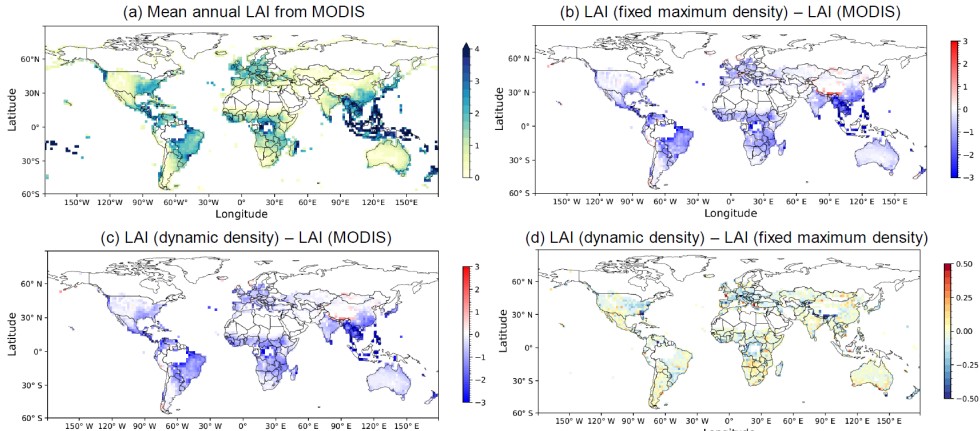

**Figure 8.** Comparison of simulated and observed mean annual LAI for grasslands averaged from 2004 to 2020. (**a**) Observed mean annual LAI from MODIS. Differences in mean annual LAI are shown for: (**b**) fixed maximum density approach minus MODIS, (**c**) dynamic density approach minus MODIS, and (**d**) dynamic density approach minus fixed maximum density approach. The figure and analyses were limited to grid cells where the sum of $V_{fra}$ values from the three grassland types exceeds 0.1.

The ratio of the relative difference in LAI to the relative difference in mortality between the two approaches was computed at the level of individual PFTs (Fig. 9). The analysis focused on grid cells where: (1) the grass density was less than unity in the dynamic density approach, and (2) mortality occurred with the fixed maximum density approach. The first condition was chosen because LAI is influenced by density only when it is below unity (according to Eq. (6)). The second condition was applied to illustrate mortality reduction in the dynamic density approach and to prevent invalid ratio calculations. A ratio lower than 1 indicates that LAI reduces very little while mortality decreases significantly, suggesting that plant productivity is allocated in a way that better supports vegetation fitness. In contrast, a ratio greater than 1 reflects a substantial decrease in LAI with only a small reduction in mortality. The results showed that 84%, 81% and 75% grid cells in temperate $C_3$, $C_4$, and tropical $C_3$ grasslands, respectively, had ratios below 1 (Fig. 9).



385

**Figure 9.** Ratio of the relative difference in LAI to the relative difference in mortality events between the fixed maximum and dynamic density approaches for temperate $C_3$ grasslands (**a**), $C_4$ grasslands (**b**), and tropical $C_3$ grasslands (**c**). The relative difference was calculated by subtracting the value in the fixed maximum density approach from that in the dynamic density approach, then dividing by the value in the fixed maximum density approach. To ensure valid calculations, both LAI and mortality values were required to be greater than zero in the fixed maximum density approach. Mortality events were accumulated over a 17-year period for this analysis, and LAI values were averaged over the same time span.



## 4 Discussion

### 4.1 The implementation of dynamic grass density

This study introduces a novel approach in ORCHIDEE r9010 to calculate grassland density as a trade-off between the reserve and labile carbon in individual plants and the total number of plants. This differs from other models that simulate a dynamic grassland density, such as the spatial patch dynamics model PATCHMOD (Wu et al., 1994) and the individual and process-based grassland model GRASSMIND (Wirth et al., 2021), which simulate more explicit demographic processes. Our simplified yet effective approach allows dynamic grass density to be simulated at the typical scale of a grid cell in the land surface model ORCHIDEE, i.e., between 50 km × 50 km and 200 km × 200 km, allowing for a global and computationally efficient representation of grassland dynamics. Most importantly, the simulated grass density varies across regions and grassland PFTs, consistent with the wide range of grassland densities reported in the literature for areas such as the Great Plains, western France, and Mongolian Plateau (e.g., Vogel et al., 2001; Dusseux et al., 2014; John et al., 2018).

Although the simulated density aligns with literature-based values, a direct comparison with global observations remains challenging due to incomplete datasets. Grass density is known to be strongly influenced by resource availability, particularly water (Schneider et al., 2008; John et al., 2018). Our results demonstrate that ORCHIDEE effectively captures a positive relationship between precipitation and simulated grass density, with density declining notably under low annual precipitation. For ORCHIDEE, this relationship is an emerging property, as a relationship between precipitation and grassland density was not coded as such. It can therefore be concluded that the proposed approach is able to simulate grassland density as the outcome of essential processes such as competition, survival, and mortality under varying resource availability (Deblauwe et al., 2008).

Additionally, the precipitation-density relationship is also a valuable diagnostic indicator to identify and evaluate potential biases in model behaviour. In some cases, the model maintained maximum grassland density despite scarce precipitation. For instance, while $C_4$ grasses are known for their resilience to extreme conditions (Taylor et al., 2010), our model simulates their density at maximum levels even when precipitation falls below 1 mm per day—an overly optimistic outcome. As part of this study, we have already recalibrated the dynamic density approach—with a focus on $C_4$ grasslands in Southern Africa—to increase the model's sensitivity to low precipitation. Applying this recalibration globally leads to a generally improved performance, with the model capturing a plausible emergent relationship between precipitation and density in most grasslands. Despite this improvement, the aforementioned counterintuitive result is not entirely eliminated. Understanding the conditions enabling this result, and potentially developing more adaptive parameterizations, will therefore be a key focus for future model investigations.

The use of the fraction of vegetation type ($V_{fra}$) as a proxy for evaluating the grass density also introduces uncertainties. If the land cover map is accurate, high $V_{fra}$ could be interpreted as an indicator of favourable climatic conditions for a given PFT. Low $V_{fra}$ may reflect unfavourable climatic conditions, but it may also be attributed to non-climatic factors, which are not considered in this study, such as fire, grazing, and human management (Chang et al., 2016; Chang et al., 2021). In addition, the land cover map may overlook subpixel vegetation structure of grasslands. For example, an area with a homogeneous mixture of grass and bare soil may be classified entirely as grassland with a high $V_{fra}$, even though the actual grassland density might be lower due to the sparse vegetation. Conversely, in locations where a remote sensing product can resolve distinct patches of grass and bare soil, only the grass-covered areas may be identified as grassland, while adjacent bare soil is classified as separate bare soil. In such cases, grass density may be high, but $V_{fra}$ appears low. This could also account for the cases where PFTs with negligible $V_{fra}$ still exhibited substantial grass density. It highlights the importance of considering bare soil distribution in the classification of grassland PFTs from land cover maps, particularly when interpreting or validating grass density.





Future work should focus on further refining water stress parameters and reserve and labile carbon targets to better capture: (1) the response of individual grasslands to climate and (2) the density of co-existing grassland PFTs. Additionally, incorporating the effects of agricultural practices and disturbances could further improve the capability of ORCHIDEE in simulating the geographical distribution and density of grasslands.

**4.2 Reduced mortality events with the dynamic density approach**

The vegetation in semi-arid regions, where extreme conditions are unfavourable for growth, tends to have low productivity and is prone to mortality events (Fig. S7; Wang et al., 2022). In ORCHIDEE, carbon starvation could result in grassland mortality during lasting droughts. At the beginning of the next growing season, the model establishes a new grassland. However, if mortality events are frequent (in this study assumed as five or more occurrences over 51 simulation years), it

suggests that the grasslands are not viable in ORCHIDEE, which contradicts the vegetation map indicating grassland present at that location. Therefore, frequent mortality events of grassland might indicate a shortcoming in either the model or the PFT map. In the dynamic density approach, more grassland grid cells were able to maintain certain productivity for growth compared to the fixed maximum density approach (Fig. S7), resulting in a significant reduction in mortality events over grasslands.

To separate model limitations from shortcomings in the PFT map, the analysis was complemented by MODIS LAI data. If the LAI value at a pixel is lower than 0.1, the land cover map may inaccurately reflect the presence of grassland, and the mortality of the prescribed vegetation is expected. In contrast, for pixels where MODIS LAI exceeds 0.1, grasslands are expected to survive in the corresponding ORCHIDEE grid cell, and the mortality in ORCHIDEE should be infrequent and primarily drought-driven. This makes these events a key diagnostic for model limitations. The remaining occurrence of mortality events

with the dynamic density approach suggests that in semi-arid regions, simulated plant resistance to water stress might be underestimated during drought, leading to a more rapid collapse once soil moisture falls below the wilting point. As a result, ORCHIDEE could trigger mortality events earlier than is observed in reality. Furthermore, adjusting photosynthetic capacity, stomatal conductance and the wilting point as a function of the growing environment (Chebbo et al., 2025) may better reflect plant resilience to prolonged water stress. Since ORCHIDEE model uses fixed parameter values within a PFT, a possible

improvement could be to calculate photosynthetic capacity, stomatal conductance and the wilting point as a function of long-term mean precipitation, thereby distinguishing between dryer and wetter climatic regions. Such parameters would then reflect the different grass species that grow under different climate conditions. This would be consistent with the evidence that perennial grasses can modify their drought-tolerant traits to enhance survival under extreme dry conditions (Norton et al., 2016; Guo et al., 2017). For example, species like *Danthonia californica*, *Geranium dissectum* and *Alopecurus pratensis* dominate

in wet regions, while *Lupinus bicolor*, *Bromus hordeaceus* and *Cenchrus ciliaris* are common in dry regions (Gubsch et al., 2011; Sandel et al., 2011; Kattge et al., 2020). The former tends to have a lower maximum carboxylation rate but higher stomatal conductance than the latter (Prentice et al., 2014).

**4.3 The aridity threshold for frequent mortality events in grasslands**

To facilitate the analysis in this study, "frequent mortality" is defined as occurring five times or more over a 51-year simulation

period, based on ecological records of widespread, drought-driven mortality in perennial grasslands. While this provides a reasonable benchmark, we acknowledge that it is a simplified assumption. Future research could further evaluate the robustness of our findings across a range of threshold values and ecosystems.

Given the same high aridity, grasslands tend to die less frequently in the dynamic density approach compared to the fixed maximum density approach. This property was not coded as such but emerged from the model's dynamics, suggesting that

adjusting vegetation density is an effective strategy for adapting to difficult growing conditions. This is consistent with the



observed spatial self-organisation in grasslands that has been explained as an adaptation to water availability (Rietkerk et al., 2002).

Aridities of 0.54, 0.69, and 0.83 have been found to result in vegetation decline, soil degradation, and systemic breakdown respectively (Berdugo et al. 2020). With the dynamic density approach, frequent mortality events became apparent for an aridity exceeding 0.9, a threshold similar to that reported by Berdugo et al. (2020). The dynamic density approach thus reproduced a realistic response of grasslands to aridity. Moreover, if the thresholds reported by Berdugo et al. (2020) are globally valid, this could suggest an inconsistency in the land cover maps used by ORCHIDEE where 60% of the grasslands are prescribed at locations with an aridity index exceeding 0.83. At these locations, the mortality of the grassland PFTs might be a realistic representation of grassland ecology. In arid regions, drought-adapted species such as succulents (Buckland et al., 2023), halophytes (Hussain et al., 2023) and phreatophytes (Sommer et al., 2014) are expected instead of grasses.

### 4.4 Implications of grassland LAI in ORCHIDEE

With the dynamic density approach, simulated mean annual LAI is expected to be lower than that in the fixed maximum density approach. This is because, according to Eq. (6), LAI is a function of both the leaf biomass of an individual and the number of individuals. Our study demonstrated that a majority of the grid cells had a lower mean annual LAI when dynamic grass density was accounted for (Fig. 8d). There is a small proportion of grid cells showing higher simulated LAI values with the dynamic density approach, which may be attributed to a trade-off effect (Jongejans et al., 2006). On one hand, grass density is reduced to alleviate growth stress caused by limited resource availability (Harper, 1977). On the other hand, decreased plant density in resource-limited conditions (e.g., water, nutrients) alleviates competition among plants and improves individual biomass growth (Springer, 2021). Although grass density is lower, the individual leaf biomass increases due to improved growing conditions. As a consequence, this trade-off may explain the slightly higher LAI with the dynamic density approach for several grid cells.

The underestimation of grassland LAI in ORCHIDEE compared to MODIS was not fully resolved with the dynamic density approach. From the MODIS data perspective, the derivation of the LAI product using a global-scale process model may affect its accuracy and lead to an overestimation of LAI by approximately 2–15% (Fensholt et al., 2004). On the ORCHIDEE side, hydrological processes are a key factor; the underestimation can be partially explained by the runoff-to-precipitation ratio (Critchley et al., 2013) and the representation of evapotranspiration (Burchard-Levine et al., 2021) under water-limited conditions. Furthermore, uncertainties in key biophysical parameters, such as the Specific Leaf Area (SLA) used to convert leaf biomass to LAI, may also constrain simulated leaf growth. Addressing this issue likely requires a multi-faceted strategy; future work should therefore focus on improving surface hydrology representation, refining plant functional parameters via data assimilation, and expanding model evaluation using a broader set of remote sensing products.

This study goes beyond a direct comparison of simulated and observed LAI by exploring how observed LAI can inform on the likelihood that grassland is present at a given location. While land cover maps may include classification uncertainties, incorporating observational LAI—particularly when LAI exceeds 0.1—improves the reliability of mortality event analyses. In the dynamic density approach, the low frequency of mortality in regions with observed LAI from MODIS > 0.1 suggests that grasslands can persist by adjusting their density to current climatic conditions. As a result, the need for artificial re-establishment by the model is greatly reduced. In contrast, the fixed maximum density assumption leads to frequent mortality events in some regions, followed by a substantial influx of carbon from the biomass of the re-established grassland. As a result, model outputs may suggest unrealistically high productivity solely due to the frequent need to re-establish the vegetation. By comparison, the dynamic density approach ensures the survival of the grassland community by allowing for density reduction, at the cost of potentially lower productivity and LAI values compared to the fixed maximum density approach. However, the




ratio between the relative reductions in LAI and mortality (Fig. 9) indicates a beneficial trade-off, where the reduction in LAI is less pronounced than the reduction in mortality events for the majority of grid cells.

### 4.5 Consequences for PFT maps

Poulter et al. (2015) provide a cross-walking table that helps assign land cover classes from remote sensing products into PFTs used in ORCHIDEE. However, such an approach may introduce inconsistencies, as the dynamic density approach in this study simulates a bare soil fraction within grasslands based on grassland density. The areas classified as bare soil may in reality contain sparse or seasonal vegetation that should be represented by a grassland or shrubland PFT, especially in semi-arid systems. Future model implementations could improve realism by replacing the fixed "bare soil" PFT with a dynamic representation of bare ground. This modification would also reduce potential biases in vegetation dynamics and mortality

events in ORCHIDEE, especially in dryland ecosystems where vegetation is sparse but not entirely absent.

### 5 Conclusions and future perspectives

The default grassland representation in ORCHIDEE, which assumes a fixed maximum density, does not adequately reflect the sparse and low-density vegetation typically found in resource-limited regions, such as semi-arid areas. This mismatch— between the model's imposed high density and the limited ecological capacity to sustain such density under low resource

availability—results in frequent and unrealistic mortality events. Therefore, this study proposed a simple yet effective approach to simulate the dynamic grassland density in the land surface model ORCHIDEE. The core of this approach is to link grassland density directly to the plant's carbon status, which serves as an integrated indicator of vegetation vitality. Specifically, grass density decreases when reserve and labile carbon in the plant are insufficient, while it increases when reserve, labile and fruit carbon are abundant.

The implementation of dynamic grass density in ORCHIDEE led to several notable improvements: (1) Grass density is now simulated within the range of 0.05–1 globally, positively correlating with annual precipitation as an emerging property. (2) Mortality events are substantially reduced compared to the default approach in ORCHIDEE. (3) Incorporating dynamic grass density raises the aridity thresholds for frequent mortality events in grasslands, with this behaviour emerging from the new approach rather than being explicitly prescribed. (4) While the dynamic density approach produces lower mean annual LAI

values than the fixed maximum density approach, it maintains a realistic level of ecosystem productivity and dramatically reduces grassland mortality, thereby enhancing the model's ecological realism. (5) The dynamic density approach eliminates the need to assign arbitrary bare soil fractions when constructing PFT maps, as the bare soil fraction emerges dynamically from the simulation, thereby improving the realism and consistency of land cover representation.

Furthermore, the improvements pave the way for estimating the dust emission from bare soil in semi-arid grasslands.

Integrating this capability will enhance dust emission estimates and provide a more comprehensive understanding of land–atmosphere feedbacks in Earth System Models.

This study also reveals several areas where grassland simulations in ORCHIDEE could be improved: (1) Refining the phenological processes and parameters for $C_4$ grasslands, including adjustments to reserve and labile carbon targets, as well as optimization of parameters in the water stress function. (2) Developing a more robust mechanism for triggering mortality

events in grasslands, especially under extreme climate conditions. (3) Enhancing vegetation classification in ORCHIDEE by incorporating more accurate vegetation maps, and explicitly representing sparse grasslands or mixed bare soil and vegetation areas in semi-arid regions, except in regions that are fully bare such as deserts.



*Code and data availability.* The ORCHIDEE model is open source and licensed under the CeCILL (CEA CNRS INRIA Logiciel Libre). The specific ORCHIDEE r9010 code used in this study is archived on Zenodo and accessible via: https://doi.org/10.5281/zenodo.15723740 (Xu, 2025a). The code to process data and generate the Figs. 3–9 in this study is archived on Zenodo at: https://doi.org/10.5281/zenodo.15877635 (Xu, 2025b).

The MODIS grasslands LAI data can be obtained from https://doi.org/10.5067/MODIS/MCD15A3H.061 (Myneni et al., 2021), and Sentinel-2 grasslands LAI can be obtained from https://www.environment.snu.ac.kr/s2lai (Wan et al., 2024). CRUJRA

data are available from https://catalogue.ceda.ac.uk/uuid/aed8e269513f446fb1b5d2512bb387ad/ (Harris et al., 2020). The PFT map in ORCHIDEE is based on the ESA CCI Land Cover database (https://www.esa-landcover-cci.org/), and the details of the initial data processing for ORCHIDEE are available at https://orchidas.lsce.ipsl.fr/dev/lccci/.

*Author contributions.* S.L. proposed the research topic. S.X. implemented the dynamic density approach in ORCHIDEE r9010 and performed the simulations and analysis. Y.B. supervised the execution. S.L. and N.V. provided expertise in ORCHIDEE. S.L., Y.B., P.C., and N.V. offered scientific guidance for the analysis and interpretation of the results. N.V. provided MODIS grassland LAI data, and L.W. provided Sentinel-2 grassland LAI data. S.X. wrote the paper, and all authors contributed to commenting and writing the manuscript.

*Competing interests.* The authors declare that they have no conflict of interest.

*Acknowledgements.* S.X. acknowledges the support of the ORCHIDEE development team, and the discussions with Fabienne Maignan and Camille Abadie. The simulations benefited from obelix, the computing cluster of LSCE.

*Financial support.* This project has received state aid from the National Research Agency (Agence Nationale de la Recherche) under the France 2030 program, with the reference ANR-22-EXTR-0009, and the funding from the European Union's Horizon Europe research and innovation program under Grant Agreement N° 101071247 (Edu4Climate – European Higher Education Institutions Network for Climate and Atmospheric Sciences).

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
