# Peer review of "Representing dynamic grass density in the land surface model ORCHIDEE r9010"

_EGUsphere, 2025_

## Author Comment (AC1)

**Referee #1**

**Comment A1**

**Overall assessment**

This manuscript presents a physiology-based dynamic grass density approach for the ORCHIDEE land surface model that addresses key limitations of the fixed-density representation.

Here is this reviewer's understanding: By linking vegetation density to reserve and labile carbon (C) pools, the model adjusts dynamically to resource availability. This mechanism reduces unrealistic mortality events, produces a more realistic emergent slope between precipitation and density, and generates the bare soil fraction directly rather than prescribing it. Together, these advances increase ecological realism, provide a basis for dust emission modelling and improve IPSL-CM performance across major grassland biomes. The study appears timely, well designed and methodologically sound, but several revisions that could strengthen its impact.

**Response**

We sincerely thank the reviewer for the insightful and positive comments that outline our main findings. We have thoroughly revised the manuscript based on the valuable suggestions, and outlined below the point-by-point responses.

**Comment A2**

**Validation and ecological realism**

Validation relies mainly on indirect proxies such as LAI and precipitation correlations. Including regional case studies that use field-based estimates of grass density or bare soil cover would allow for more direct evaluation and strengthen confidence in the model's realism.

**Response A2**

We have followed the reviewer's suggestion to strengthen the validation of our model. We performed an analysis comparing simulated grassland density with field-based estimates over five representative regions: a temperate European grassland (France), the Eurasian steppes (Mongolia), a North American meadow (USA), a Sahelian rangeland (Senegal), and a semi-arid grass—shrub community (Australia).

Although the metrics from the field-based observation are not identical to the grassland density defined in our study, to mitigate this gap, we have selected the five case studies (Booth et al., 2005; Dusseux et al., 2014; John et al., 2018; Melville et al., 2019; Diatta et al., 2023) that provide metrics conceptually similar to our definition of density: the fractional area occupied by conceptual individuals.

The results from this comparison are summarized in Table 1. The simulated annual mean grass densities show an overall good agreement with field observations, supporting the

ecological realism of the model. For example, in France, observed value for grassland density range from 0.91 to 0.99, while the model simulated 0.95; similar consistency was found in the United States (0.68 observed vs 0.63 simulated) and Australia (0.10–0.60 observed vs 0.15 and 0.50 simulated). In Senegal, the simulated value of 0.18 remains near the lower bound of the observed range (0.06 to 0.79). In Mongolia, the different steppe types (typical, meadow, and desert) represent plot-based locations. This presents a scale mismatch when comparing them to the coarse spatial resolution in ORCHIDEE, but the results are still in agreement.

Details of this new evaluation and its rationale have been added (lines 183–197) to the new section "2.3 Model evaluation against regional field observations and global dataset" in Methods, as:

"In order to directly assess the ecological realism of the simulated grassland density, we compared model outputs with field-based estimates from five published regional case studies. These studies span a range of grassland ecosystems: a temperate European grassland in France (Dusseux et al., 2014), the Eurasian steppe on the Mongolian Plateau (John et al., 2018), a meadow in the USA (Booth et al., 2005), a Sahelian rangeland in Senegal (Diatta et al., 2023), and a grass-shrub community in Australia (Melville et al., 2019), as listed in Table

We acknowledge that the metrics from field-based observation are not identical to the grassland density defined in our study. However, the five case studies provide metrics that are thought to be sufficiently similar to be compared to the metric in ORCHIDEE, i.e., the fractional area occupied by conceptual individuals (Fig. 1a–b). The case-studies provide the area-based geometric estimates—either by counting points classified as vegetation within quadrats (John et al., 2018; Diatta et al., 2023), along transects (Booth et al., 2005; Melville et al., 2019), or from downward-facing hemispherical photographs to estimate green vegetation cover (Dusseux et al., 2014). Detailed descriptions of each dataset, including observed and corresponding simulated values, measurement methods, and caveats of the selected methods, are provided in Table 1. The hemispherical photography method may be influenced by plant height and leaf area (Dusseux et al., 2014); the effects of grazing were controlled by selecting fenced sites (Diatta et al., 2023); and the observational sites included not only grasses but also forbs and shrubs, although grasses were dominant (Melville et al., 2019)."

The full results interpretation has been added to the Results subsection in section "3.2 Evaluation of simulated grassland density" (lines 366–375), as:

"The simulated grassland density was compared against direct field-based estimates for five regional case studies (Table 1). Over temperate grassland in France, the simulated density of 0.95 was within the observed range of 0.91 to 0.99 (Dusseux et al., 2014). This consistency extended to the Upper Beaver Meadows site in North America, with a simulated density of 0.63 that approached the observed mean of 0.68 (Booth et al., 2005). For the desert steppe (with the cold desert climate) of the Mongolian Plateau, the simulated value of 0.27 was just outside the observed range of 0.10–0.26 (John et al., 2018). Furthermore, simulated average densities for typical steppes characterized by the semi-arid climate (0.40) and meadow steppes characterized by the subarctic climate (0.63) fell within their respective observed ranges of 0.34–0.50 and 0.45–0.78 (John et al., 2018). In the Sahelian fenced rangeland of

Senegal, the simulated density of 0.18 was in the low range of the large observed range of 0.06 to 0.79. Finally, for the mixed grass-shrub community in Australia, both the simulated  $C_4$  (0.15) and tropical  $C_3$  (0.50) grass densities were consistent with the field-based range of 0.1 to 0.6 (Melville et al., 2019)."

The discussion of strengths and limitations was included in the section "4.1 The implementation of dynamic grassland density" (lines 515–529), as:

"The evaluation against five case studies (Table 1) gives confidence in the model's ability to represent grassland density across different grass PFTs and locations. The close agreement at all the five sites suggests our model accurately captures the central tendency of grassland density. Despite these encouraging results, this evaluation should be interpreted with caution due to several key uncertainties. The primary challenge is the conceptual mismatch between our simulated "density" and the observational metrics. The mismatch was mitigated by selecting the closest available conceptual analogues (Sect. 2.3). However, the discrepancies cannot be fully eliminated. For example, in the Australian grass-shrub community (Melville et al., 2019), the field-based metric unavoidably includes shrubs, thus resulting in higher values compared to a pure grassland ecosystem. While the close agreement (Table 1) suggests the dynamic density approach captured the dominant grass trend, the shrublands in Australia might also be misclassified as grasslands in the PFT maps in ORCHIDEE, which would lead to our model simulating grasslands in the shrub-contaminated areas. This alignment may therefore stem partly from this PFT misclassification. In addition, the scale mismatch between plot-level field data and the model's coarse grid-cell resolution is another source of uncertainty, particularly in heterogeneous landscapes like the Mongolian Plateau. Despite this spatial discrepancy, the result that our simulated value range aligned with the observed range suggests the new approach captures the ecological gradient across different steppes: with higher values in meadow steppe, medium values in typical steppe, and lower values in desert steppe (Booth et al., 2005; Dusseux et al., 2014; John et al., 2018; Melville et al., 2019; Diatta et al., 2023)."

**Below is the new Table 1 added in the manuscript:**

**Table 1.** Evaluation of simulated grassland density from ORCHIDEE against field-based estimates from various grassland sites (all values in m2 m-2).

| Site/Region | Observed Value   | Simulated Value | Observational Method and Caveats               | Model Value Extraction                     |
|-------------|------------------|-----------------|------------------------------------------------|--------------------------------------------|
| Yar         | 0.91-0.99        | 0.95            | Fraction of vegetation cover from              | Temperate C 3 grassland density |
| Watershed,  |                  |                 | downward-facing hemispherical                  | extracted at 3° W, 47° N.                  |
| France      |                  |                 | photographs taken approximately 1 m            |                                            |
|             |                  |                 | above the canopy (Dusseux et al., 2014).       |                                            |
|             |                  |                 | Caveat : The observed value is affected |                                            |
|             |                  |                 | by plant height and leaf area, which           |                                            |
|             |                  |                 | might influence the consistency with           |                                            |
|             |                  |                 | grassland density.                             |                                            |
| Mongolian   | 0.45-0.78        | 0.63±0.35       | Canopy cover from grid-square                  | Temperate C 3 grassland density |
| Plateau     |                  |                 | counting, measured by counting the             | extracted for each steppe type.            |
| (meadow     |                  |                 | number of 10×10 grid mesh filled with          | See Note* for coordinates.                 |
| steppe)     |                  |                 | vegetation within a 0.5×0.5m quadrat           |                                            |
| Mongolian   | 0.34–0.5         | 0.40±0.24       | (John et al., 2018).                           |                                            |
| Plateau     |                  |                 |                                                |                                            |
| (typical    |                  |                 |                                                |                                            |
| steppe)     | 0.4.0.04         | 0.05            |                                                |                                            |
| Mongolian   | 0.1–0.26         | 0.27±0.06       |                                                |                                            |
| Plateau     |                  |                 |                                                |                                            |
| (desert     |                  |                 |                                                |                                            |
| steppe)     |                  |                 |                                                |                                            |
| The Upper   | 0.68 (0.52–0.86) | 0.63            | Green cover from point-intercept               | Temperate C 3 grassland density |
| Beaver      |                  |                 | transects, classifying a functional group      | extracted at 105° W, 39° N.                |

| Meadows,
USA              |           |                                                            | (green vegetation or bare ground) at points spaced every 30 cm along two parallel 50-meter transects (for a total of 166 points per transect) by a two-member crew (Booth et al., 2005).                                                                        |                                                                                                |
|------------------------------|-----------|------------------------------------------------------------|-----------------------------------------------------------------------------------------------------------------------------------------------------------------------------------------------------------------------------------------------------------------|------------------------------------------------------------------------------------------------|
| Ferlo,
Senegal            | 0.06–0.79 | 0.18                                                       | Visual estimation of vegetation coverage in 1 m 2 quadrats. Selected the ungrazed, fenced site (Diatta et al., 2023).  Caveat: Data is from a fenced, ungrazed site to exclude grazing effects.                                                      | The C 4 grassland density extracted at 15° W, 15° N.                                |
| Fowlers
Gap,
Australia | 0.1–0.6   | 0.15 (C 4 );
0.50 (tropical C 3 ) | Photosynthetic vegetation fraction from star transects, by recording every meter along three 100-meter tapes laid out in a star pattern (Melville et al., 2019).  Caveat: The field site is a mixed community of grasses, forbs and shrubs, not pure grassland. | The C 4 and tropical C 3 grassland densities extracted at 141° E, 31° S. |

\*Note: According to Figure 1 in John et al. (2018), we delineated three types of steppe on the Mongolian Plateau in ORCHIDEE: 97° E–103° E, 45° N–47° N in the meadow steppe, excluding other steppe types within this rectangle; 111° E–117° E, 39°N–47°N in the typical steppe, excluding forest meadow and meadow steppe within this range; 89°E–111°E, 39°N–45°N in the desert steppe, excluding desert and typical steppe areas.

**References:**

Booth, D. T., Cox, S. E., Fifield, C., et al.: Image analysis compared with other methods for measuring ground cover, Arid Land Res. Manag., 19, 91–100, https://doi.org/10.1080/15324980590916486, 2005.

Diatta, O., Ngom, D., Ndiaye, O., Diatta, S., and Taugourdeau, S.: Structure and phenology of herbaceous stratum in the Sahelian rangelands of Senegal, Grasses, 2, 98–111, https://doi.org/10.3390/grasses2020009, 2023.

Dusseux, P., Vertès, F., Corpetti, T., et al.: Agricultural practices in grasslands detected by spatial remote sensing, Environ. Monit. Assess., 186, 8249–8265, https://doi.org/10.1007/s10661-014-4001-5, 2014.

John, R., Chen, J., Giannico, V., et al.: Grassland canopy cover and aboveground biomass in Mongolia and Inner Mongolia: Spatiotemporal estimates and controlling factors, Remote Sens. Environ., 213, 34-48, https://dx.doi.org/10.1016/j.rse.2018.05.002, 2018.

Melville, B., Fisher, A., and Lucieer, A.: Ultra-high spatial resolution fractional vegetation cover from unmanned aerial multispectral imagery, Int. J. Appl. Earth Obs. Geoinf., 78, 14–24, https://doi.org/10.1016/j.jag.2019.01.013, 2019

**Comment A3**

The mortality—recruitment scheme, based on C pool trade-offs, is elegant in its simplicity but assumes asexual recruitment. Explicitly discussing the limitations of this assumption would help readers understand how the approach may underperform in ecosystems dominated by seed banks or sexual reproduction.

**Response A3**

Thank you very much for this insightful comment. We agree that the assumption of asexual recruitment is a simplification that warrants discussion. Following the advice, we have added a paragraph in section 4.1 to explicitly explain this assumption and its limitations, particularly

for ecosystems driven by sexual reproduction or seed banks. Lines 501-506 consist of the following new text:

"In ORCHIDEE, the recruitment scheme is represented as asexual recruitment, based on the assumption that grasslands are dominated by perennial species. Most perennial grasses primarily reproduce asexually through clonal stems derived from belowground tissues, while sexual reproduction via seeds plays a comparatively smaller role (Blair et al., 2013). In contrast, annual plants rely exclusively on seeds for yearly regeneration. While our model's assumption captures the dominant strategy in perennial grasslands, we acknowledge it as a limitation: the model may underperform in ecosystems where sexual reproduction and persistent seed banks are the primary drivers of recruitment."

**Reference:**

Blair, J., Nippert, J., and Briggs, J.: Grassland ecology, in: Ecology and the Environment, edited by: Monson, R., Springer, New York, 389–423, https://doi.org/10.1007/978-1-4614-7612-2 14, 2013.

**Comment A4**

Parameter recalibration, particularly for C4 grasslands, improves outcomes, yet the robustness of these changes remains uncertain. Providing an additional sensitivity analysis, for example in supplementary material, to show how density responds to parameter variation would bolster confidence in the results.

**Response A4**

We thank the referee for this valuable suggestion to expand the sensitivity analysis of the C4 grassland parameters. As suggested, we have made the following revisions.

We performed additional sensitivity analyses for two key parameter sets and have synthesized the results in Fig. S5 and Fig. S6 for southern Africa. We would like to point out that, during this revision, we identified and corrected an issue in the previous version of Figure S6. We have ensured that the revised figure now accurately reflects the simulated density.

Regarding carbon target scaling factor, we ran two additional simulations with scaling factors of 0.75 (Fig. S5b) and 1.25 (Fig. S5d). Following the suggestion of the reviewer, new panel (Fig. S5f) has been added to synthesize the relationship between grassland density and the range of scaling factors (0.5, 0.75, 1.0, 1.25, and 1.5). As shown in Fig. S5f, the average grassland density (diamonds) remained greater than 0.9 and relatively insensitive across most factors, with a notable exception of showing a slight drop in response to the 1.5 scaling factor. In contrast, the spatial variability (represented by the 5th\_95th percentile range) was highly sensitive. This range was narrow for factors of 0.5 and 0.75, but widened dramatically for values greater than 1. This widening, particularly at 1.5, was driven by a significant drop in the 5th percentile, indicating much greater spatial heterogeneity and that a larger portion of grid cells was experiencing lower density.

Regarding water stress formulation, we added a new panel (Figure S6f) to systematically compare the impact of different water stress formulations (linear vs. exponential with  $\alpha = 1$ ,

2, 4, and 8) on grassland density. This new panel (Fig. S6f) revealed a non-linear response to the formulation change. The model was relatively insensitive to the choice between the linear formulation and exponential formulations with low  $\alpha$  values (e.g.,  $\alpha$ =1, 2). In these cases, both the mean density and the 5th\_95th percentile range remained high and stable, indicating uniformly high grassland density. The impact became pronounced at higher  $\alpha$  values. At  $\alpha$ =4, the percentile range began to widen (driven by a drop in the 5th percentile), indicating an increase in spatial heterogeneity. This effect was strongest at  $\alpha$ =8, where both the mean density and the 5th percentile dropped significantly. This resulted in the widest variability range, reflecting the much lower densities seen in the corresponding spatial map (Fig. S6e).

We have revised the section "2.8 Tuning of C4 grassland parameters", to include a detailed analysis of these new results, explaining how grassland density responds to the different parameter sets. The new text can be found in lines 307–314, and lines 325–331.

Lines 307–314: "As shown in Fig. S5f, this analysis revealed that the average density (diamonds) over this region remained high (greater than 0.9) and relatively insensitive across most factors, with only a slight drop for a scaling factor of 1.5. In contrast, the spatial variability (represented by the 5th–95th percentile range) was more sensitive to the scaling factor. This range was narrow for factors of 0.5 and 0.75 but widened significantly for values greater than 1. This widening, particularly at 1.5, was driven by a significant drop in the 5th percentile, indicating much greater spatial heterogeneity because a larger portion of grid cells was experiencing lower density (as also seen in Fig. S5e). Although a scaling factor of 1.5 slightly decreased the regional mean, it introduced a spatial variability that better reflected real-world heterogeneity. Therefore, a value of 1.5 was applied to increase the target level for reserve and labile carbon in C4 grasslands."

Lines 325–331: "As shown in Fig. 6f, the model was relatively insensitive to the choice between the linear formulation and exponential formulations for low  $\alpha$  values (e.g.,  $\alpha=1,2$ ). In these cases, both the mean density and the 5th–95th percentile range remained high and stable, indicating uniformly high grassland density. The impact became pronounced at higher  $\alpha$  values. At  $\alpha=4$ , the percentile range began to widen (driven by a drop in the 5th percentile), indicating an increase in spatial heterogeneity. This effect was strongest at  $\alpha=8$ , where both the mean density and the 5th percentile dropped significantly. This latter setting resulted in the widest variability range, reflecting the much lower densities seen in the corresponding spatial map (Fig. S6e). Therefore,  $\alpha=8$  was selected for the global simulations to enhance water stress sensitivity of C4 grasslands."

Below are the updated Fig. S5 and Fig. S6:

**Figure S5.** Grassland density (averaged from 2004 to 2020) in southern Africa  $C_4$  grasslands in the dynamic density approach with different scaling factor for the reserve and labile carbon target. (a–e) The scaling factor was chosen as 0.5, 0.75, 1, 1.25 and 1.5. (f) The relationship between the scaling factor and grassland density, plotting the mean value across all pixels (diamonds) and the  $5^{th}$ – $95^{th}$  percentile range (shaded area).

**Figure S6.** Grassland density (averaged from 2004 to 2020) in southern Africa C4 grasslands in the dynamic density approach with alternative water stress formulations. (a–e) Spatial distribution of grassland density under a linear water stress formulation by default (a), and an exponential formulation with the parameter α set to 1 (b), 2 (c), 4 (d) and 8 (e). (f) Grassland density as a function of the water stress formulation, showing the mean value across all pixels (diamonds) and the 5th–95th percentile range (shaded area).

**Comment A5**

**External uncertainties and broader implications**

The paper also acknowledges uncertainties in prescribed plant functional type (PTF) maps, including the unrealistic placement of grasses in hyper-arid zones. Quantifying the extent to which such mapping errors contribute to remaining mortality artefacts would help distinguish external sources of error from limitations internal to the model.

**Response A5**

We thank the reviewer for highlighting the potential influence of prescribed PFT maps on mortality artefacts.

To quantify this, we conducted a targeted spatial analysis. First, we identified all grassland grid cells where mortality events occurred in the simulation with dynamic density approach (coloured points in Fig. S2). We then screened these locations for potential PFT map errors using three criteria, identifying grid cells mapped as "grassland" but which were unsuitable for survival:

- (1) Location in hyper-arid regions: The grid cell was in a hyper-arid region (Aridity Index  $\leq$  0.03, according to Zomer et al., 2022), where vascular plants are typically restricted to ephemeral streams (Huang et al., 2016; Groner et al., 2023).
- (2) Low observed LAI: The observed MODIS LAI was below 0.1, whereas viable grasslands typically exhibit the LAI greater than 0.1 (Si et al., 2012; Haynes et al., 2019).
- (3) High aridity: The calculated aridity exceeded 0.83, a threshold implying ecosystem breakdown (Berdugo et al., 2020).

Grid cells meeting any of these criteria were classified as "constrained regions" unsuitable for grassland and marked in red (Fig. S2). We then calculated the fraction of mortality cells occurring within these constrained regions for all grassland PFTs (temperate C3, C4, and tropical C3). These constrained cells (red points) accounted for 97% of all grassland mortality grid cells (Fig. S2).

This analysis allows us to distinguish one specific external data error—the potential incorrect classification of grasslands in constrained regions—from internal model limitations. The finding that this specific error accounts for 97% of total mortality suggests that the majority of mortality events is likely to originate from the potential misclassification of PFT maps.

This targeted analysis and its discussion have been incorporated into the revised manuscript in the Methods (Section 2.5) in lines 251–260:

"To quantify the impact of PFT mapping errors on simulated grassland mortality, we first identified all grassland grid cells where mortality events occurred in the simulation using the dynamic density approach (Fig. S2). Next, a set of criteria was established to identify "constrained regions" where the persistence of grassland vegetation is considered unlikely. A grid cell was classified as constrained if it met at least one of the following three conditions: (1) Location within a hyper-arid zone: In these zones, little vegetation can survive, and vascular plants are often restricted to ephemeral streams receiving runoff (Huang et al., 2016; Groner et al., 2023). (2) Critically low LAI: The observed LAI for viable grasslands is typically greater than 0.1 (Si et al., 2012; Haynes et al., 2019), which suggests regions with mean annual LAI

**Figure S2.** Spatial distribution of simulated grassland mortality artefacts. Grey squares denote grid cells where grassland mortality events occur in the simulations, while red squares indicate those located in constrained regions (hyper-arid regions, critically low LAI, or ecosystem breakdown) where grassland PFTs are unrealistically prescribed.

**Reference:**

Berdugo, M., Delgado-Baquerizo, M., Soliveres, S., et al.: Global ecosystem thresholds driven by aridity, Science, 367, 787–790, https://doi.org/10.1126/science.aay5958, 2020.

Groner, E., Babad, A., Berda Swiderski, N., et al.: Toward an extreme world: The hyper-arid ecosystem as a natural model, Ecosphere, 14, e4586, https://doi.org/10.1002/ecs2.4586, 2023.

Haynes, K. D., Baker, I. T., Denning, A. S., et al.: Representing grasslands using dynamic prognostic phenology based on biological growth stages: Part 2. Carbon cycling, J. Adv. Model. Earth Syst., 11, 4440–4465, https://doi.org/10.1029/2018MS001540, 2019.

Huang, J., Ji, M., Xie, Y., et al.: Global semi-arid climate change over last 60 years, Climate Dynamics, 46, 1131–1150, https://doi.org/10.1007/s00382-015-2636-8, 2016.

Si, Y., Schlerf, M., Zurita-Milla, R., et al.: Mapping spatio-temporal variation of grassland quantity and quality using MERIS data and the PROSAIL model, Remote Sens. Environ., 121, 415–425, https://doi.org/10.1016/j.rse.2012.02.011, 2012.

Zomer, R. J., Xu, J., and Trabucco, A.: Version 3 of the Global Aridity Index and Potential Evapotranspiration Database, Sci. Data, 9, 409, https://doi.org/10.1038/s41597-022-01493-1, 2022.

**Comment A6**

Although dust flux simulations are planned for future work, the manuscript would benefit from a conceptual schematic linking dynamic density, emergent bare soil fractions and dust emission potential. Such a figure would highlight the broader significance of the study.

**Response A6**

We thank the referee for this constructive suggestion. We have added a new conceptual schematic (Figure 1c, as shown below) to illustrate the links between dynamic grassland density, emergent bare soil fractions, and dust emission potential.

**Figure 1.** Conceptual framework of grassland density under varying resource availability and its link to dust emission. With high resource availability, grassland density is able to reach the maximum density (a), while low resource availability dynamically results in lower grassland density (b). The conceptual framework (c) illustrates the mechanism linking vegetation dynamics to dust emission. The schematic shows how climatic drivers control dynamic grassland density, which in turn determines the bare soil fraction and surface erodibility. Dust emission is triggered when the surface is exposed to sufficient wind erosivity, creating a potential feedback loop with the climate system.

**Comment A7**

**Presentation and minor issues**

Presentation could be improved through more consistent terminology, particularly in distinguishing "density" from "cover" and in clarifying what constitutes "an individual" in the model. Although the manuscript explains that "density" differs from "plant cover", it sometimes uses "density" in a way that resembles "cover", e.g. in the statement "... whereas grassland density reflects grass and bare soil fractions within the grassland PFT" (Line 285), which conflicts with the earlier definition of "the number of individuals per unit area" (line 60).

**Response A7**

We thank the reviewer for this valuable comment and for pointing out the inconsistency in our use of the term "density". To address this fundamental point, we have implemented a series of systematic revisions centred on the formal introduction of the term "conceptual individual".

First and mostly importantly, we have improved the explanatory paragraph at the first mention of "grassland density" (now in lines 64–70) that distinguishes "density" from "cover" and clarifies that an "individual" refers to the conceptual unit used in the model. This new text is designed to provide readers with a clear framework from the very beginning:

"In this study, we focus on population density, defined as the number of individuals per unit area. Here, each individual represents a conceptual unit that occupies 1 m² of land, rather than a physical plant. Accordingly, the unit of grassland density in this study is expressed as m² per m². For instance, a hectare of grassland with a density of 1 contains 10,000 individuals, occupying a total area of 10,000 m² per hectare (Fig. 1a). A density of 0.25 therefore corresponds to 2,500 individuals occupying 2,500 m² per hectare (Fig. 1b). In this framework, grassland density thus relates to the geometrically fractional occupancy of conceptual individuals, and differs from "plant cover" which refers to the optically projected vegetation coverage in grasslands."

With this foundational definition in place, we then revised other specific sections of the manuscript for clarity and consistency.

As the reviewer suggested, we have revised the following sentence in lines 385–386 (formerly line 285) to align with our model's framework. The revised sentence now reads:

"... whereas grassland density reflects the fractional area occupied by conceptual individuals within the grassland PFT"

This revision clarifies that "density" in our study refers specifically to the fractional occupancy by these conceptual units, thereby clearly distinguishing it from the concept of "plant cover" and resolving the conflict.

We also reinforced the definition in the Methods section (now lines 114-116), highlighting that our density variable, D, is based on these "conceptual" units:

"... where D refers to grassland density, defined as the fractional area occupied by conceptual individuals (m2 m-2). By default, the number of conceptual individuals ( $N_{\text{max}}$ ) in grassland is set to be 10,000 per hectare, with each occupying 1 m2 of land. Consequently, the default vegetation density for grasslands in the model is fixed at 1 m2 m-2."

Finally, we have performed a thorough review of the entire manuscript to ensure the term "conceptual individual" is applied consistently, removing any potential for ambiguity.

**Comment A8**

Likewise, the methods section states that "each individual is assumed to occupy 1 m2" (lines 104-105), yet discussions of biomass allocation and asexual reproduction obscure the line between a biological plant and an abstract unit, potentially confusing readers. For example, in "This approach for increasing grassland density reflects grass recruitment through asexual means, which is a suitable method for representing perennial plants" (lines 153-155), it should be clarified that the "individual" is a conceptual unit, not a physical plant.

**Response A8**

We agree with the reviewer that the distinction between a biological plant and the abstract unit used in the model was not sufficiently clear.

As the reviewer suggested, we clarified the sentence discussing asexual reproduction to explicitly connect the biological process to our modelling approach using conceptual units (now lines 172–173, formerly lines 153–155):

"This approach for increasing grassland density reflects asexual recruitment of perennial plants (Blair et al., 2013), which is implemented in the model using conceptual units rather than actual plants."

**Comment A9**

The distinction between vegetation type fraction - "a value for its fraction ( $V_{\rm fra}$ ), line 91 - and "density" is also sometimes unclear, with "density" referring to surface coverage rather than actual counts of individuals., e.g. "... land cover map represents the fraction of vegetation type ( $V_{\rm fra}$ ) for each PFT within one grid cell, whereas grassland density represents grass and bare soil fractions within the grassland PFT" (lines 284–285).

**Response A9**

We thank the reviewer for pointing out this lack of clarity. We agree that the distinction between the vegetation type fraction ( $V_{\rm fra}$ ) and density needs to be sharpened. To create a clear and direct comparison, we have revised the sentence in lines 384–386 (formerly lines 284–285) as follows:

"... the land cover map represents the fractional area covered by each PFT ( $V_{\rm fra}$ ) within one grid cell, whereas grassland density reflects the fractional area occupied by conceptual individuals within the grassland PFT."

This revision now explicitly defines the two terms in relation to scale:  $V_{\rm fra}$  refers to the fractional cover at the grid-cell level, while the density describes the fractional occupancy within the PFT's designated area.

**Comment A10**

A schematic showing C redistribution during density adjustments would help readers follow the mechanism, and adding explicit mortality thresholds to figure annotations (e.g. Fig. 7) would improve interpretability.

**Response A10**

We thank the reviewer for the suggestions, which have helped improve the clarity and readability of our figures.

1. In response to the suggestion "A schematic showing C redistribution during density adjustments would help readers follow the mechanism", we have added a new schematic figure (Fig. 2) to illustrate this process. The figure and its caption are presented below.

Figure 2. Schematic of the carbon (C) redistribution mechanism during density adjustments. The model simulates the transition from an initial state with density  $D_1$  (top) to two possible scenarios after adjustment to density  $D_2$  (bottom): a decrease in density (a) or an increase in density (b). Blue indicates an increase and red indicates a decrease in values for both carbon pools (rectangles) and grassland density (circles).

2. In response to the suggestion "adding explicit mortality thresholds to figure annotations (e.g. Fig. 7) would improve interpretability", we have added a dashed line to Fig. 7 in the revised manuscript. We have also revised the caption to explicitly define the threshold in Fig. 7, as shown below.

Figure 7. Relationship between aridity and mortality events over three types of grassland. Panels ( $\mathbf{a}$ - $\mathbf{c}$ ) show the relationship using the fixed density approach, while panels ( $\mathbf{d}$ - $\mathbf{f}$ ) show it using the dynamic density approach. The grassland types are temperate  $C_3$  ( $\mathbf{a}$ ,  $\mathbf{d}$ ),  $C_4$  ( $\mathbf{b}$ ,  $\mathbf{e}$ ), and tropical  $C_3$  ( $\mathbf{c}$ ,  $\mathbf{f}$ ). The mortality events were accumulated over 51 simulation years, and the aridity was calculated for the same period. The dashed line at five mortality events marks the threshold, separating "infrequent mortality" from more frequent events.

While implementing this change, we also took the opportunity to re-examine the figure's underlying data. We identified that we previously used an older variable for potential evapotranspiration (evapot) using an old method. We have now updated this to the more recent variable (evapot\_corr) provided by ORCHIDEE to calculate aridity. This methodological update improves the accuracy of the figure's analysis (mainly affects the fixed density approach) and strengthens our findings. We have also updated the relevant text and values throughout the manuscript to ensure they are consistent with the corrected Fig. 7.

**Comment A11**

Minor grammatical polishing would further smooth the narrative. For example, awkward phrasing, such as "... the mortality in ORCHIDEE should be infrequent and primarily ..." (line 443) would flow better as "... mortality in ORCHIDEE should occur infrequently and mainly ...", or "... grassland dies in the ORCHIDEE model and ..." (lines 174-175) would be better if worded as "... the grassland is considered dead in ORCHIDEE, and ...".

**Response A11**

We thank the reviewer for the valuable suggestions to improve the manuscript's readability. We agree with the proposed changes and have revised the sentences accordingly.

The sentence in lines 588–589 (formerly line 443) has been rephrased as: "... mortality in ORCHIDEE should occur infrequently and be mainly driven by drought."

The sentence in lines 229–230 (formerly lines 174–175) has been rephrased as: "... the grassland is considered dead in ORCHIDEE, and ..."

**Comment A12**

Using the simple present tense to model descriptions would also enhance the writing, e.g. changing "Adding to these limitations, a fixed density fails to respond to changes in resource availability, hindering the possibility of studying the response of dust emissions ..." (lines 71-72) to "In addition, a fixed density does not respond to resource availability, which hinders the study of dust emission responses ...".

**Response A12**

We thank the reviewer for this constructive suggestion on improving our writing style.

Following this advice, we have revised the sentence in lines 80–81 (formerly lines 71–72). The sentence now reads:

"In addition, a fixed density does not respond to resource availability, which hinders the study of dust emission responses ..."

We have also performed a thorough review of the manuscript to ensure that model descriptions consistently use the simple present tense where appropriate.

**Comment A13**

Removing phrases such as "Note that" and "including" from "Note that the carbon of other compartments (including leaf, aboveground stem, root and fruit) in each individual remains ..." (lines 124-125) would allow for the following: "The carbon in other compartments (leaf, stem, root, fruit) remains ...". Likewise, "Both of the events ..." (line 191) could simply be shortened to "The events ...".

**Response A13**

We thank the reviewer for the suggestions on making our phrasing more concise. We have adopted these recommendations as follows:

The sentence in lines 141–142 (formerly 124–125) has been revised by removing "Note that" and "including" as suggested. It now reads:

"The carbon in other compartments (leaf, stem, root and fruit) in each conceptual individual remains ..."

For consistency, we also applied this revision to a similar sentence in lines 173–174, which now reads:

"The carbon in other compartments (leaf, stem and root) in each conceptual individual remains constant."

The sentence in line 247 (formerly line 191) has been shortened to "The events ..." as recommended.

**Comment A14**

Finally, unit notation should follow SI conventions, with spaces before unit symbols and negative exponents for "per" relationships. For example, "gC m-2 per day" should be written as "g C m-2 d-1", denoting grams of carbon per square meter per day. Likewise, the unit "m2 gC-1" is ambiguous and could be misread as "square meters times grams per carbon". To remove this confusion, it should be rewritten as "m2 g-1 C", which distinctly indicates square meters per gram of carbon.

**Response A14**

We appreciate the reviewer's valuable suggestion regarding unit notation.

The unit of "gC m-2 per day" has been revised to "g C m-2 d-1" in line 167, and the unit of "m2 gC-1" has been changed to "m2 g-1 C" in line 163.

Throughout the manuscript, we have ensured that all units conform to SI conventions by inserting spaces before unit symbols (e.g., "g C") and using negative exponents to express "per" relationships. For example, "gC per individual" has been consistently revised to "g C ind-1". The same approach has been applied to other similar units across the text.

**Comment A15**

**Summary and recommendation**

This study represents a significant methodological advance for ORCHIDEE and makes an important contribution to Earth system modelling. Strengthening validation, clarifying demographic simplifications and refining presentation would further enhance its impact. With these minor revisions, the manuscript will be a valuable and timely addition.

**Response A15**

We sincerely appreciate the reviewer's positive and encouraging assessment of our study. As detailed in our point-by-point responses above, we have thoroughly addressed all recommendations regarding model validation, clarification of demographic simplifications, and refinement of presentation, and have incorporated the corresponding revisions into the manuscript.

We are grateful for the reviewer's constructive and supportive feedback.

---

## Author Comment (AC2)

**Referee #2**

**Comment B1**

This manuscript presents a new implementation of a dynamic grassland density scheme within the ORCHIDEE land surface model. The proposed approach allows grass density to vary in response to physiological carbon reserves, enabling a more flexible and ecologically realistic representation of vegetation cover, particularly under semi-arid conditions. The overall scheme is well justified. The authors evaluate this scheme using multiple lines of evidence, including relationships between precipitation and grass density, frequency of grassland mortality events, and comparisons with satellite-derived LAI products. The proposed dynamic density scheme effectively mitigates key limitations of the original model in simulating grassland dynamics under semi-arid conditions, thereby offering substantial value for model development and holding considerable potential for broader scientific impact. The manuscript is well written. However, the current version of the manuscript requires some improvements in the rigor and comprehensiveness of the model evaluation.

**Response**

We thank the reviewer for the positive assessment of our work and for the constructive feedback regarding the need to enhance the rigor and comprehensiveness of the model evaluation. We appreciate the constructive criticism as it will strengthen this aspect of the manuscript. In the revised version, we have carefully addressed this point, as detailed in our point-by-point responses below.

**Comment B2**

(1) The evaluation relies primarily on indirect indicators (e.g., LAI, mortality frequency) without sufficient direct evidence that the model accurately reproduces observed spatial patterns of grass density or vegetation coverage. Comparison against datasets such as vegetation coverage (e.g., vegetation fractional coverage data) may provide insights in the model improvements?

**Response**

We thank the reviewer for this constructive suggestion. As recommended, we sought a direct comparison against a fractional vegetation cover dataset to test for the model's ability to represent spatial patterns.

Following this recommendation, we have performed a rigorous evaluation for the simulated fractional vegetation cover (FVC) against the Copernicus Land Monitoring Service FCOVER dataset (Copernicus Land Monitoring Service, 2020). We selected the year 2004 for this comparison, as it matches the static global land cover map used throughout this study. The FCOVER product (originally at ~0.003° resolution) was regridded to our model's  $2^{\circ}\times2^{\circ}$  resolution. To ensure a fair comparison, we:

1. Calculated the corresponding fractional vegetation cover (*FVC*) specifically from the targeted grassland PFTs within ORCHIDEE using the equation:

$$FVC = D_{\text{temp C3}} \times V_{\text{fra,temp C3}} + D_{\text{C4}} \times V_{\text{fra,C4}} + D_{\text{trop C3}} \times V_{\text{fra,trop C3}}$$

where  $D_{\text{temp C3}}$ ,  $D_{\text{C4}}$ , and  $D_{\text{trop C3}}$  are the simulated grassland density (1.0 for the fixed density approach, 0.05–1.0 for our new dynamic density approach) and  $V_{\text{fra,temp C3}}$ ,  $V_{\text{fra,C4}}$ , and  $V_{\text{fra,trop C3}}$  are the fractional area of each grassland PFT (temperate C3, C4, tropical C3) in one grid cell.

2. Applied a (semi-)arid region mask (based on Zomer et al., 2022) to focus the analysis on our target ecosystems where grasslands dominate and exclude the canopy cover from other vegetation as much as possible.

The results of this direct comparison (Fig. S7) illustrated the improvements brought by the new dynamic approach. Regarding the spatial patterns, the spatial correlation (Pearson's r) between the model and the FCOVER dataset increased from a r=0.11 with the old approach to r=0.24 with our new (dynamic) approach. The new approach also achieved a lower RMSE (0.22) compared to the old approach (0.26).

This improvement was obvious over the western United States, Asia, southern Africa, and Australia (Fig. S7), where the new dynamic scheme simulated a lower and more realistic *FVC*, in closer agreement with the dataset, compared to the fixed density approach.

**Figure S7.** Fraction of vegetation cover from FCOVER product (a), simulations with fixed density approach (b) and dynamic density approach (c) in 2004.

It should be noted that two main caveats apply to this comparison, which can explain the remaining deviation from observations: (1) In (semi-)arid regions, the FCOVER product includes all green vegetation (e.g., shrubs, crops), whereas our calculation here includes only the grassland PFTs that we improved. (2) The current model version does not yet account for disturbances like grazing or fire (Chang et al., 2016; Chang et al., 2021), which are known to reduce *FVC* and are implicitly included in the satellite observations.

However, the fact that our new approach achieved a clear relative improvement: the spatial correlation (Pearson's r) with the FCOVER data increased from 0.11 (with the old approach) to 0.24 (with our new dynamic approach), alongside a 15% reduction in RMSE, despite these known model and data differences. It thus provides the evidence that our new dynamic density scheme is a substantial step toward greater ecological realism.

The corresponding revisions regarding the direct comparison of *FVC* were implemented in the Methods (lines 204–216):

"Furthermore, the simulated fractional vegetation cover was compared against the Copernicus Land Monitoring Service FCOVER product (Copernicus Land Monitoring Service, 2020). We selected the year 2004 for this comparison, as it matches the static global land cover map used throughout this study. The FCOVER product (originally at ~0.003° resolution) was regridded to our model's 2°×2° resolution using RemapCon (Jones, 1998; Goudiaby et al., 2024) in the Climate Data Operators library for Linux. To ensure a fair comparison, we calculated the corresponding fractional vegetation cover (*FVC*) specifically from the targeted grassland PFTs within ORCHIDEE using the equation:

$$FVC = D_{\text{temp C3}} \times V_{\text{fra,temp C3}} + D_{\text{C4}} \times V_{\text{fra,C4}} + D_{\text{trop C3}} \times V_{\text{fra,trop C3}}$$

$$\tag{6}$$

where  $D_{\text{temp C3}}$ ,  $D_{\text{C4}}$ , and  $D_{\text{trop C3}}$  are the simulated grassland density (1.0 for the fixed density approach, 0.05–1.0 for our new dynamic density approach) and  $V_{\text{fra,temp C3}}$ ,  $V_{\text{fra,C4}}$ , and  $V_{\text{fra,trop C3}}$  are the fractional area of each grassland PFT (temperate C3, C4, tropical C3) within one grid cell.

Given that this study aims to improve grassland density simulation, the comparison of *FVC* focused specifically on grasslands. To isolate the target ecosystems where grasslands dominate and exclude the canopy cover from other vegetation as much as possible, we applied a (semi-)arid region mask based on the aridity index map by Zomer et al. (2022)."

**Results (lines 376–383):**

"The dynamic density approach was further evaluated against a comparison of FVC with a global satellite-based FCOVER product (Copernicus Land Monitoring Service, 2020) (Fig. S7). The spatial correlation (Pearson's r) between the model and the FCOVER data increased from r = 0.11 with the fixed density approach to r = 0.24 with the dynamic density approach. The dynamic density approach also exhibited a lower RMSE (0.22) compared to the fixed density approach (0.26). This improvement was particularly evident in western United States, Asia, southern Africa, and Australia, where the dynamic scheme simulated a lower and more realistic FVC (Fig. S7c), in better agreement with the FCOVER dataset, compared to the fixed density approach (Fig. S7b). Such regional-scale improvement is consistent with the findings from the regional field-based comparisons."

**Discussion (lines 530–537):**

"As shown in the results (Sect. 3.2), the direct *FVC* comparison against the FCOVER satellite product (Copernicus Land Monitoring Service, 2020) also supported the new dynamic approach, which improved both spatial correlation (r) and RMSE. There are two main caveats in this comparison, which likely explain the deviation from observations: (1) In (semi-)arid regions, the FCOVER product includes all green vegetation (e.g., shrubs, crops), whereas our calculation was focused only on the grassland PFTs we improved. (2) The current model does not yet account for key disturbances like grazing or fire, which are known to affect *FVC* and

are implicitly included in the satellite observations (Chang et al., 2016; Chang et al., 2021). Nevertheless, the fact that our new scheme showed a clear improvement despite these known mismatches underscores the robustness of the new dynamic density approach."

In addition, to further enhance the rigor and comprehensiveness of our evaluation, we have also added five new regional case studies comparing the model against field-based observations, covering diverse ecosystems, including a temperate European grassland (France), the Eurasian steppes (Mongolia), a North American meadow (USA), a Sahelian rangeland (Senegal), and a semi-arid grass—shrub community (Australia).

Although the metrics from the field-based observation are not identical to the grassland density defined in our study, to mitigate this gap, we have selected the five case studies (Booth et al., 2005; Dusseux et al., 2014; John et al., 2018; Melville et al., 2019; Diatta et al., 2023) that provide metrics conceptually similar to our definition of density: the fractional area occupied by conceptual individuals.

The results from this comparison are summarized in Table 1. The simulated annual mean grass densities show an overall good agreement with field observations, supporting the ecological realism of the model. For example, in France, observed value for grassland density range from 0.91 to 0.99, while the model simulated 0.95; similar consistency was found in the United States (0.68 observed vs 0.63 simulated) and Australia (0.10–0.60 observed vs 0.15 and 0.50 simulated). In Senegal, the simulated value of 0.18 remains near the lower bound of the observed range (0.06 to 0.79). In Mongolia, the different steppe types (typical, meadow, and desert) represent plot-based locations. This presents a scale mismatch when comparing them to the coarse spatial resolution in ORCHIDEE, but the results are still in agreement.

Details of this new evaluation and its rationale have been added (lines 183–197) to the new section "2.3 Model evaluation against regional field observations and global dataset" in Methods, as:

"In order to directly assess the ecological realism of the simulated grassland density, we compared model outputs with field-based estimates from five published regional case studies. These studies span a range of grassland ecosystems: a temperate European grassland in France (Dusseux et al., 2014), the Eurasian steppe on the Mongolian Plateau (John et al., 2018), a meadow in the USA (Booth et al., 2005), a Sahelian rangeland in Senegal (Diatta et al., 2023), and a grass-shrub community in Australia (Melville et al., 2019), as listed in Table 1.

We acknowledge that the metrics from field-based observation are not identical to the grassland density defined in our study. However, the five case studies provide metrics that are thought to be sufficiently similar to be compared to the metric in ORCHIDEE, i.e., the fractional area occupied by conceptual individuals (Fig. 1a–b). The case-studies provide the area-based geometric estimates—either by counting points classified as vegetation within quadrats (John et al., 2018; Diatta et al., 2023), along transects (Booth et al., 2005; Melville et al., 2019), or from downward-facing hemispherical photographs to estimate green vegetation cover (Dusseux et al., 2014). Detailed descriptions of each dataset, including observed and corresponding simulated values, measurement methods, and caveats of the selected methods, are provided in Table 1. The hemispherical photography method may be

influenced by plant height and leaf area (Dusseux et al., 2014); the effects of grazing were controlled by selecting fenced sites (Diatta et al., 2023); and the observational sites included not only grasses but also forbs and shrubs, although grasses were dominant (Melville et al., 2019)."

The full results interpretation has been added to the Results subsection in section "3.2 Evaluation of simulated grassland density" (lines 366–375), as:

"The simulated grassland density was compared against direct field-based estimates for five regional case studies (Table 1). Over temperate grassland in France, the simulated density of 0.95 was within the observed range of 0.91 to 0.99 (Dusseux et al., 2014). This consistency extended to the Upper Beaver Meadows site in North America, with a simulated density of 0.63 that approached the observed mean of 0.68 (Booth et al., 2005). For the desert steppe (with the cold desert climate) of the Mongolian Plateau, the simulated value of 0.27 was just outside the observed range of 0.10–0.26 (John et al., 2018). Furthermore, simulated average densities for typical steppes characterized by the semi-arid climate (0.40) and meadow steppes characterized by the subarctic climate (0.63) fell within their respective observed ranges of 0.34–0.50 and 0.45–0.78 (John et al., 2018). In the Sahelian fenced rangeland of Senegal, the simulated density of 0.18 was in the low range of the large observed range of 0.06 to 0.79. Finally, for the mixed grass-shrub community in Australia, both the simulated C4 (0.15) and tropical C3 (0.50) grass densities were consistent with the field-based range of 0.1 to 0.6 (Melville et al., 2019)."

The discussion of strengths and limitations was included in the section "4.1 The implementation of dynamic grassland density" (lines 515–529), as:

"The evaluation against five case studies (Table 1) gives confidence in the model's ability to represent grassland density across different grass PFTs and locations. The close agreement at all the five sites suggests our model accurately captures the central tendency of grassland density. Despite these encouraging results, this evaluation should be interpreted with caution due to several key uncertainties. The primary challenge is the conceptual mismatch between our simulated "density" and the observational metrics. The mismatch was mitigated by selecting the closest available conceptual analogues (Sect. 2.3). However, the discrepancies cannot be fully eliminated. For example, in the Australian grass-shrub community (Melville et al., 2019), the field-based metric unavoidably includes shrubs, thus resulting in higher values compared to a pure grassland ecosystem. While the close agreement (Table 1) suggests the dynamic density approach captured the dominant grass trend, the shrublands in Australia might also be misclassified as grasslands in the PFT maps in ORCHIDEE, which would lead to our model simulating grasslands in the shrub-contaminated areas. This alignment may therefore stem partly from this PFT misclassification. In addition, the scale mismatch between plot-level field data and the model's coarse grid-cell resolution is another source of uncertainty, particularly in heterogeneous landscapes like the Mongolian Plateau. Despite this spatial discrepancy, the result that our simulated value range aligned with the observed range suggests the new approach captures the ecological gradient across different steppes: with higher values in meadow steppe, medium values in typical steppe, and lower values in desert steppe (Booth et al., 2005; Dusseux et al., 2014; John et al., 2018; Melville et al., 2019; Diatta et al., 2023)."

Below is the new Table 1 added in the manuscript:

**Table 1.** Evaluation of simulated grassland density from ORCHIDEE against field-based estimates from various grassland sites (all values in m2 m-2).

| Site/Region                                 | Observed Value   | Simulated Value                                            | Observational Method and Caveats                                                                                                                                                                                                                                                 | Model Value Extraction                                                                                |
|---------------------------------------------|------------------|------------------------------------------------------------|----------------------------------------------------------------------------------------------------------------------------------------------------------------------------------------------------------------------------------------------------------------------------------|-------------------------------------------------------------------------------------------------------|
| Yar
Watershed,
France                 | 0.91–0.99        | 0.95                                                       | Fraction of vegetation cover from downward-facing hemispherical photographs taken approximately 1 m above the canopy (Dusseux et al., 2014). Caveat: The observed value is affected by plant height and leaf area, which might influence the consistency with grassland density. | Temperate C 3 grassland density extracted at 3° W, 47° N.                                  |
| Mongolian
Plateau
(meadow
steppe)  | 0.45–0.78        | 0.63±0.35                                                  | Canopy cover from grid-square counting, measured by counting the number of 10×10 grid mesh filled with vegetation within a 0.5×0.5m quadrat                                                                                                                                      | Temperate C 3 grassland density extracted for each steppe type. See Note* for coordinates. |
| Mongolian
Plateau
(typical
steppe) | 0.34–0.5         | 0.40±0.24                                                  | (John et al., 2018).                                                                                                                                                                                                                                                             |                                                                                                       |
| Mongolian
Plateau
(desert
steppe)  | 0.1–0.26         | 0.27±0.06                                                  |                                                                                                                                                                                                                                                                                  |                                                                                                       |
| The Upper
Beaver
Meadows,
USA      | 0.68 (0.52–0.86) | 0.63                                                       | Green cover from point-intercept transects, classifying a functional group (green vegetation or bare ground) at points spaced every 30 cm along two parallel 50-meter transects (for a total of 166 points per transect) by a two-member crew (Booth et al., 2005).              | Temperate C 3 grassland density extracted at 105° W, 39° N.                                |
| Ferlo,
Senegal                           | 0.06–0.79        | 0.18                                                       | Visual estimation of vegetation coverage in 1 m 2 quadrats. Selected the ungrazed, fenced site (Diatta et al., 2023).  Caveat: Data is from a fenced, ungrazed site to exclude grazing effects.                                                                       | The C 4 grassland density extracted at 15° W, 15° N.                                       |
| Fowlers
Gap,
Australia                | 0.1-0.6          | 0.15 (C 4 );
0.50 (tropical C 3 ) | Photosynthetic vegetation fraction from star transects, by recording every meter along three 100-meter tapes laid out in a star pattern (Melville et al., 2019).  Caveat: The field site is a mixed community of grasses, forbs and shrubs, not pure grassland.                  | The C 4 and tropical C 3 grassland densities extracted at 141° E, 31° S.        |

\*Note: According to Figure 1 in John et al. (2018), we delineated three types of steppe on the Mongolian Plateau in ORCHIDEE: 97° E–103° E, 45° N–47° N in the meadow steppe, excluding other steppe types within this rectangle; 111° E–117° E, 39°N–47°N in the typical steppe, excluding forest meadow and meadow steppe within this range; 89°E–111°E, 39°N–45°N in the desert steppe, excluding desert and typical steppe areas.

**References:**

Booth, D. T., Cox, S. E., Fifield, C., et al.: Image analysis compared with other methods for measuring ground cover, Arid Land Res. Manag., 19, 91–100, https://doi.org/10.1080/15324980590916486, 2005.

Chang, J., Ciais, P., Gasser, T., et al.: Climate warming from managed grasslands cancels the cooling effect of carbon sinks in sparsely grazed and natural grasslands, Nat. Commun., 12, 118, https://doi.org/10.1038/s41467-020-20406-7, 2021.

Chang, J., Ciais, P., Herrero, M., et al.: Combining livestock production information in a process-based vegetation model to reconstruct the history of grassland management, Biogeosciences, 13, 3757–3776, https://doi.org/10.5194/bg-13-3757-2016, 2016.

Copernicus Land Monitoring Service: Fraction of Green Vegetation Cover 2014-present (raster 300 m), global, 10-daily—version 1. Copernicus Land Monitoring Service [Data set]. <a href="https://doi.org/10.2909/09578c73-4f5d-4d2c-90ff-4e17fb7dbf69">https://doi.org/10.2909/09578c73-4f5d-4d2c-90ff-4e17fb7dbf69</a>, 2020 (last access: 01/11/2025).

Diatta, O., Ngom, D., Ndiaye, O., Diatta, S., and Taugourdeau, S.: Structure and phenology of herbaceous stratum in the Sahelian rangelands of Senegal, Grasses, 2, 98–111, https://doi.org/10.3390/grasses2020009, 2023.

Dusseux, P., Vertès, F., Corpetti, T., et al.: Agricultural practices in grasslands detected by spatial remote sensing, Environ. Monit. Assess., 186, 8249–8265, https://doi.org/10.1007/s10661-014-4001-5, 2014.

John, R., Chen, J., Giannico, V., et al.: Grassland canopy cover and aboveground biomass in Mongolia and Inner Mongolia: Spatiotemporal estimates and controlling factors, Remote Sens. Environ., 213, 34-48, https://dx.doi.org/10.1016/j.rse.2018.05.002, 2018.

Melville, B., Fisher, A., and Lucieer, A.: Ultra-high spatial resolution fractional vegetation cover from unmanned aerial multispectral imagery, Int. J. Appl. Earth Obs. Geoinf., 78, 14–24, https://doi.org/10.1016/j.jag.2019.01.013, 2019.

Zomer, R. J., Xu, J., and Trabucco, A.: Version 3 of the Global Aridity Index and Potential Evapotranspiration Database, Sci. Data, 9, 409, https://doi.org/10.1038/s41597-022-01493-1, 2022.

**Comment B3**

(2) The simulated LAI was compared with MODIS and Sentinel-2 LAI. However, it does not convincingly show how the dynamic density scheme improves the LAI simulation. The differences in LAI between the dynamic and fixed approaches are illustrated (e.g., Fig. 8), while there is little quantitative assessment of the improvement. The figures do not clearly highlight regions where the new scheme reduces model—data mismatches. Without clearer metrics or spatial diagnostics, the added value of the new scheme remains ambiguous. Furthermore, it is not clear whether the seasonality of LAI is improved due to the new scheme.

**Response**

We thank the reviewer for this critical and constructive feedback. We agree that a comprehensive and clear quantitative assessment of LAI is essential to demonstrate the added value of the new scheme. We have conducted a new, three-part quantitative analysis specifically designed to evaluate the improvements in LAI simulation against MODIS dataset. To more accurately assess the model's improvement in critical areas, all subsequent analyses applied a mask for (semi-)arid regions based on the aridity index map (Zomer et al., 2022). This allows us to focus on the scheme's performance in these key water-stressed environments.

First, we conducted a global-scale comparison of the mean annual grassland LAI simulated from both the old (fixed density) and new (dynamic density) approaches against the MODIS dataset. The statistics confirm a consistent, albeit modest, improvement with the new scheme: the Pearson's correlation (r) increased from 0.51 to 0.56, and the RMSE decreased from 0.60 to 0.59. This demonstrates a statistically better performance for the new approach at the global scale.

Regarding spatial diagnostics, we focused on the four representative semi-arid regions: Australia, southern Africa, Central Asia, and South America (Fig. S11a). These sites were chosen as they represent the large contiguous grassland ecosystems within the semi-arid domain on their respective continents. We found that the new dynamic scheme shows clear and consistent advantages in capturing the spatial patterns of LAI. Compared to the MODIS dataset, the coefficient of determination (R²) increased or remained unchanged in all four regions with the new approach (Fig. S11b–e, Table 2). The RMSE decreased in three of the four regions (Australia, Central Asia, and southern Africa).

Finally, we assessed the model's ability to simulate the mean seasonal cycle. The improvement is most dramatic in southern Africa (Fig. S12b). The old approach failed to capture the dry-season LAI minimum (August–October), whereas the new approach mitigates this major bias. The new dynamic density approach increased the seasonal correlation (r) with MODIS from 0.77 to 0.93, compared to the fixed density approach. However, seasonality in Australia (Fig. S12a) and South America (Fig. S12d) did not show improvements (Table 2). This suggests that other factors (e.g., processes not yet included or parameters needing optimization) may be dominant drivers of LAI seasonality in those specific regions, which helps identify clear pathways for future model development.

In summary, this quantitative assessment confirms the value of the dynamic scheme while also illuminating its limitations. The scheme's strengths are the clear, measurable improvements in global and regional spatial patterns of LAI (Fig. S11, Table 2), and the pronounced improvement in seasonal dynamics in key regions like Southern Africa (Fig. S12). Its limitations are that the global-scale improvements were modest, and the seasonal improvements were not equally significant in all regions (e.g., Australia, South America). It thus serves a crucial diagnostic purpose: it validates the new approach's effectiveness while simultaneously helping us target other key phenological processes and specific regions for further improvement. These new analyses have been added to the revised manuscript in the Results (lines 473–484):

"To refine the LAI analysis, a mask was applied to (semi-)arid regions identified by Zomer et al. (2022), focusing on water-stressed environments. Globally, compared with the MODIS dataset (Fig. 8a), the Pearson correlation coefficient (r) increased from 0.51 to 0.56, and the RMSE decreased from 0.60 to 0.59 when transitioning from the fixed density to the dynamic density approach. Spatially, statistical analysis was conducted for the four representative semi-arid regions: Australia, southern Africa, Central Asia, and South America (Fig. S11a), which were chosen as they represent the large contiguous grassland ecosystems within the semi-arid domain on their respective continents. In all four regions, the coefficient of determination (R2) improved or remained unchanged under the dynamic density approach (Fig. S11b-e, Table 2), while RMSE decreased in three regions (Australia, Central Asia, and southern Africa). Moreover, the dynamic density approach enhanced the seasonal dynamics in southern Africa (Fig. S12b, Table 2), successfully capturing the dry-season LAI minimum (August-October) that the fixed density approach failed to reproduce. The new dynamic density approach increased the seasonal correlation (r) with MODIS from 0.77 to 0.93, compared to the fixed density approach. In contrast, seasonality in Australia (Fig. S12a) and South America (Fig. S12d) did not show improvements (Table 2)."

Discussion (lines 652–657):

"The global and regional quantitative assessment against the MODIS dataset demonstrates that the dynamic density approach yields consistent, albeit modest improvements in grassland LAI (Figs. S11, S12). However, this analysis also reveals that the overall global improvement is minor, and that the issue of LAI seasonality persists. It is important to note that LAI seasonality is driven by the phenology subroutine in ORCHIDEE, which was not modified by our new dynamic density approach. Improving this phenology remains a separate, long-standing challenge in Earth System Models. This underlying issue is relevant, though, as these persistent phenological issues likely contribute to the remaining mortality events in our simulations."

The newly added Table 2, Figure S11 and Figure S12 are shown below.

**Table 2.** Statistical comparison of simulated grassland LAI (from this study) against MODIS LAI across four regions: Australia, southern Africa, Central Asia, and South America. Statistics include the coefficient of determination  $(R^2)$  and RMSE for mean annual LAI, and Pearson's r and RMSE for LAI seasonality.

| _                  | ·                      | Mean annual grasslands   | LAI                    |                          |
|--------------------|------------------------|--------------------------|------------------------|--------------------------|
| Regions            |                        | $\mathbb{R}^2$           | RMSE                   |                          |
|                    | Fixed density approach | Dynamic density approach | Fixed density approach | Dynamic density approach |
| Australia          | 0.58                   | 0.72                     | 0.39                   | 0.36                     |
| Southern
Africa | 0.13                   | 0.21                     | 0.58                   | 0.55                     |
| Central Asia       | 0.38                   | 0.40                     | 0.28                   | 0.27                     |
| South America      | 0.45                   | 0.45                     | 0.79                   | 0.81                     |
|                    |                        | LAI seasonality          |                        |                          |
| Regions            |                        | r                        | RMSE                   |                          |
|                    | Fixed density approach | Dynamic density approach | Fixed density approach | Dynamic density approach |
| Australia          | -0.60                  | -0.67                    | 0.47                   | 0.54                     |
| Southern
Africa | 0.77                   | 0.93                     | 0.14                   | 0.14                     |
| Central Asia       | 0.30                   | 0.31                     | 0.62                   | 0.62                     |
| South America      | 0.62                   | 0.60                     | 0.63                   | 0.67                     |

**Figure S11.** Comparison of simulated mean annual LAI from the fixed density and dynamic density approaches against MODIS LAI. (a) Global map of the mean annual LAI difference (Dynamic density approach – Fixed density approach). Purple boxes highlight the four representative regions: (b) Australia (113° E–155° E, 45° S–11° S), (c) southern Africa (13° E–35° E, 23° S–15° S), (d) Central Asia (41° E–119° E, 33° N–55° N), and (e) South America (75° W–45° W, 55° S–15° S). (b–e) Scatter plots comparing modelled LAI (ORCHIDEE) against observed LAI (MODIS) for each region. Red points and text correspond to the fixed density approach, while blue points and text correspond to the dynamic density approach. Statistical metrics (R², RMSE, and sample size n) are shown for each approach. The dashed black line is the 1:1 line. All values represent mean annual averages for the 2004–2020 period. The analysis for (b-e) was restricted to semi-arid and arid regions (based on the aridity index from Zomer et al., 2022) to ensure the comparison focused on grassland-dominated ecosystems. Both the "Observed LAI (MODIS)" (x-axis) and the "Modeled LAI (ORCHIDEE)" (y-axis) represent grassland LAI.

**Figure S12.** Average seasonal cycle of LAI, comparing MODIS observations with simulations from the fixed density (red line) and dynamic density (blue line) approaches. The comparison is shown for four representative regions: (a) Australia, (b) southern Africa, (c) Central Asia, and (d) South America. All data represent the mean monthly values, averaged over the 2004–2020 period. The analysis was restricted to semi-arid and arid regions (based on the aridity index from Zomer et al., 2022) to ensure the comparison focused on grassland-dominated ecosystems, where both MODIS and simulated LAI represent grassland LAI. Statistical metrics (Pearson's r and RMSE) for each approach against MODIS are shown in the corresponding colours.

**References:**

Zomer, R. J., Xu, J., and Trabucco, A.: Version 3 of the Global Aridity Index and Potential Evapotranspiration Database, Sci. Data, 9, 409, https://doi.org/10.1038/s41597-022-01493-1, 2022.

**Comment B4**

**Minor remarks:**

Figure 2 is unnecessary. The processes are quite simple and can be well understand with text only.

**Response**

Thank you for this comment. We agree that Figure 2 is unnecessary, accordingly, and we have removed this figure in the revised manuscript.